

# Multi-strategy synthetized equilibrium optimizer and application

Quandang Sun[1,2,*], Xinyu Zhang[3,*], Ruixia Jin[4], Xinming Zhang[1,3] and Yuanyuan Ma[3]

[1] Engineering Lab of Intelligence Business & Internet of Things, Xinxiang, Henan, China
[2] Henan Normal University, Software College of Software, Henan Normal University, Xinxiang, Henan, China
[3] Henan Normal University, College of Computer and Information Engineering, Xinxiang, Henan, China
[4] Sanquan College of Xinxiang Medical University, Xinxiang, Henan, China
[*] These authors contributed equally to this work.

## ABSTRACT

**Background**. Improvement on the updating equation of an algorithm is among the most improving techniques. Due to the lack of search ability, high computational complexity and poor operability of equilibrium optimizer (EO) in solving complex optimization problems, an improved EO is proposed in this article, namely the multi-strategy on updating synthetized EO (MS-EO).

**Method**. Firstly, a simplified updating strategy is adopted in EO to improve operability and reduce computational complexity. Secondly, an information sharing strategy updates the concentrations in the early iterative stage using a dynamic tuning strategy in the simplified EO to form a simplified sharing EO (SS-EO) and enhance the exploration ability. Thirdly, a migration strategy and a golden section strategy are used for a golden particle updating to construct a Golden SS-EO (GS-EO) and improve the search ability. Finally, an elite learning strategy is implemented for the worst particle updating in the late stage to form MS-EO and strengthen the exploitation ability. The strategies are embedded into EO to balance between exploration and exploitation by giving full play to their respective advantages.

**Result and Finding**. Experimental results on the complex functions from CEC2013 and CEC2017 test sets demonstrate that MS-EO outperforms EO and quite a few state-of-the-art algorithms in search ability, running speed and operability. The experimental results of feature selection on several datasets show that MS-EO also provides more advantages.

# INTRODUCTION

Traditional mathematical methods have strict requirements for optimization problems (OPs) (*Hashim et al., 2019*). They cannot get good results in solving most OPs. Inspired by natural phenomena and biological behavior (*Ghasemi et al., 2023a*), researchers have proposed many meta-heuristic algorithms (MAs) to solve OPs better. They include genetic algorithm (GA) (*Zhou et al., 2021*), simulated annealing (SA) (*Kirkpatrick, Gelatt & Vecchi, 1983*), particle swarm optimization (PSO) (*Chen & Lin, 2009*), differential evolution (DE)

Corresponding author
Ruixia Jin, jinruixia@126.com

(*Mohamed, Hadi & Jambi, 2019*), Shuffled Frog Leaping algorithm (SFLA) (*Houssein et al., 2021*), Artificial Bee Colony (ABC) (*Altay & Varol Altay, 2023*), biogeography-based optimization (BBO) (*Simon, 2008*), Cuckoo Search (CS) (*Gandomi, Yang & Alavi, 2013*), Grey Wolf Optimizer (GWO) (*Mirjalili, Mirjalili & Lewis, 2014*), *etc.* MAs are applied in many fields, such as feature selection (*Ghasemi et al., 2023b*), economic dispatch (*Ayedi, 2023*) due to their simple structure, easy application, and no derivative information on OPs. However, more and more OPs need solving urgently as modern society evolves and the OPs are more and more complex (*Liu et al., 2021*; *Ghasemi et al., 2023c*). It is very necessary to develop an MA with higher efficiency, stronger universality, and scalability. Therefore, improved MAs are constantly being proposed.

The equilibrium optimizer (EO) was proposed by *Faramarzi et al. (2020)*, and it is a new physics-based MA. EO simulates the dynamic mass balance on a control volume, that is, the conservation of mass entering, leaving, and generating. It has a unique search mechanism that updates the population by learning from the elite particles. Compared with some classical MAs such as GA, PSO, and DE, this search mechanism enables EO to obtain better performance (*Faramarzi et al., 2020*) in solving some classic OPs.

The improvement on an MA generally includes the modification of the structure, the improvement of the updating equation, and the adjustment of the parameters, *etc.* The updating equation is the core part of an algorithm. Depending on the way in which they are improved, they can simply be classified into three types.

The first is the modification of the original updating equation. According to different modification methods, it can be divided into two sub-types. (1) Simplification. The method simplifies the original updating equation by discarding some components (*Zhang et al., 2020a*),  it can reduce the computational complexity or improve the operability, but it may degenerate the performance. (2) Modification. This method alters the original updating equation to improve the performance (*Long et al., 2018*), but it may increase the parameters and the computational load due to the high complexity of the modified updating equation. The second is the addition of the updating equation. It means adding some new updating ways to the original algorithm to make up for some defects. According to different addition methods, it can be divided into 3 sub-types. (1) Individual-based addition. The method adds a new updating equation for an individual in the population, while other individuals keep the original updating method unchanged. (2) Subgroup-based addition. The method divides the population into multiple subgroups. All the agents of at least one subgroup are updated using the original updating equation while the other subgroups are updated separately using the added updating equations. (3) Iteration-based addition. The method adds a new updating equation in some iterations. Individual-based addition and subgroup-based addition can increase the possibilities of generating new solutions to more degrees and obtain more population diversity. The third is hybrid improved update methods. This means using multiple improved updating methods (*Asilian Bidgoli, Ebrahimpour-Komleh & Rahnamayan, 2020*).

Some scholars have proposed some EO variants and applied EO or its variants to some fields. *Gupta et al. (2020)* proposed an EO with a mutation strategy. It made the population maintain sufficient diversity and enhanced the global search ability of EO. *Wunnava et al.*

*(2020)* applied an adaptive EO (A-EO) to multilevel thresholding image segmentation. A-EO solved the problem that search agents were randomly scattered to nonperformer search agents in EO. *Biller et al. (2016)* proposed an improved EO through linear classification reduction diversity technique and local minima elimination method, while the variant of EO proposed by *Ghasemi et al. (2023a)*, *Ghasemi et al. (2023b)* and *Ghasemi et al. (2023c)* used different probabilities to select equilibrium candidate solutions. They were applied to photovoltaic parameter estimation, and effectively improved the optimization accuracy and reliability. These EO variants have improved the optimization performance of EO to different degrees. Although they all verify the performances of the classical functions, there are no reports on verifying the variants of EO on CEC2013 and CEC2017 test sets.

As mentioned above, modification and addition have their own shortcomings and advantages. A hybridization of the two can compensate for their respective shortcomings, and an ingenious combination of several strategies can allow individuals to work closely together to balance exploration and exploitation. Thus, the main purpose of this article is to propose many strategies based on updating equations and discuss how they can be integrated to improve the overall performance of EO and solve some of its drawbacks. Inspired by the above description, this article proposes a multi-strategy synthetic EO (MS-EO). The graphical abstract of this article is shown in Fig. 1. The contributions of this article are as follows:

- A simplified updating strategy is applied to EO to form a simplified EO (SEO) by simplifying the original updating equation to improve the operability and reduce the computational complexity.
- An information-sharing strategy is added as an updating way to SEO and two updating ways alternate in different iterations to improve the global search ability.
- A migration strategy is used to update a golden particle to get a stronger search ability.
- An elite learning strategy is implemented for the worst particle updating in the late iteration stage to enhance the local search ability.

The organization of this article is structured in the following order: the related work, the proposed algorithm (MS-EO) , the experimental results and analysis, the application of MS-EO to feature selection, conclusion and future work.

## Related work

EO is inspired by the law of dynamic mass balance. Its mathematical updating model is as follows:

$$C_i = C_e + (C_i - C_e) \otimes F + G/(\lambda \times V) \otimes (1 - F) \tag{1}$$

where $C_i$ represents the concentration of the i-th particle. $C_e$ is the concentration of the elite particle selected randomly from the elite group. $F$ is an exponential term, $G$ represents the generation rate, and $\lambda$ is the turnover rate and is often taken as a random vector with each component distributed in [0, 1]. $V$ is the volume and is usually regarded as a unit. $+$ means term by term multiplication. $i \in [1, N]$, and $N$ is the population size.

In order to maintain the diversity of the population and the direction of evolution, in the iterative process, some of the best individuals in the current population will form the

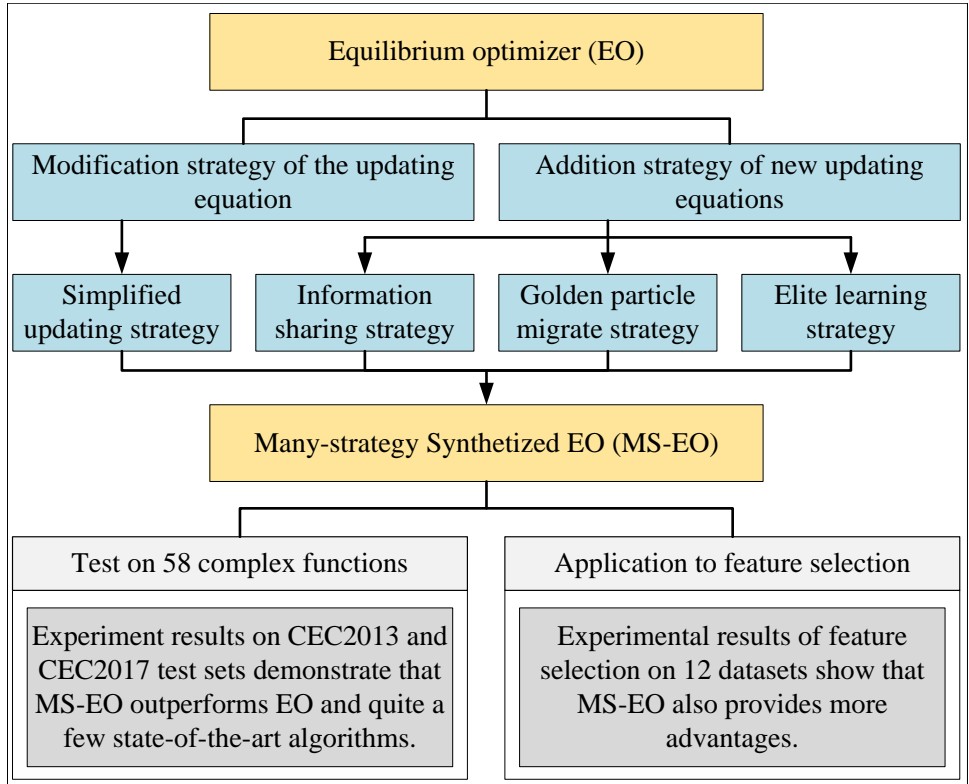

**Figure 1** The structure diagram summarizes four strategies of MS-EO (simplified updating strategy, information sharing strategy, golden particle migrate strategy and elite learning strategy), experimental setting and application.

elite group to pass to the next generation. In this article, the elite group is composed of five elite particles, which are as follows:

$$C_e \in C_\alpha, C_\beta, C_\delta, C_\gamma, C_\varphi. \tag{2}$$

The first four are the best-so-far particles' concentrations in the current population, the first $C_\alpha$, the second $C_\beta$, the third $C_\delta$ and the fourth $C_\gamma$ best one, respectively. The last particle $C_\varphi$ is the arithmetic mean of the four best particles' concentrations.

The exponential term $F$ is expressed as:

$$F = a_1 \times sign(r - 0.5) \otimes [e^{-\varepsilon \times \lambda} - 1] \tag{3}$$

where $a_1$ is a tuning parameter, *sign* is a sign function, and $r$ is a random vector in [0, 1] like $\lambda$. $\varepsilon$ is a dynamic tuning parameter and its expression is as follows:

$$\varepsilon = (1 - t/T)^{(a_2 \times t/T)} \tag{4}$$

where $t$ is the current iteration number, $T$ represents the maximum iteration number, and $a_2$ is also a parameter. $F$ is also a dynamic random parameter vector and its component value is shown in Fig. 2. Fig. 2 only illustrates the value changes on the six components of $F$ with $T$ equal to 1,000. From Fig. 2, the red line represents the value change of $F$ with

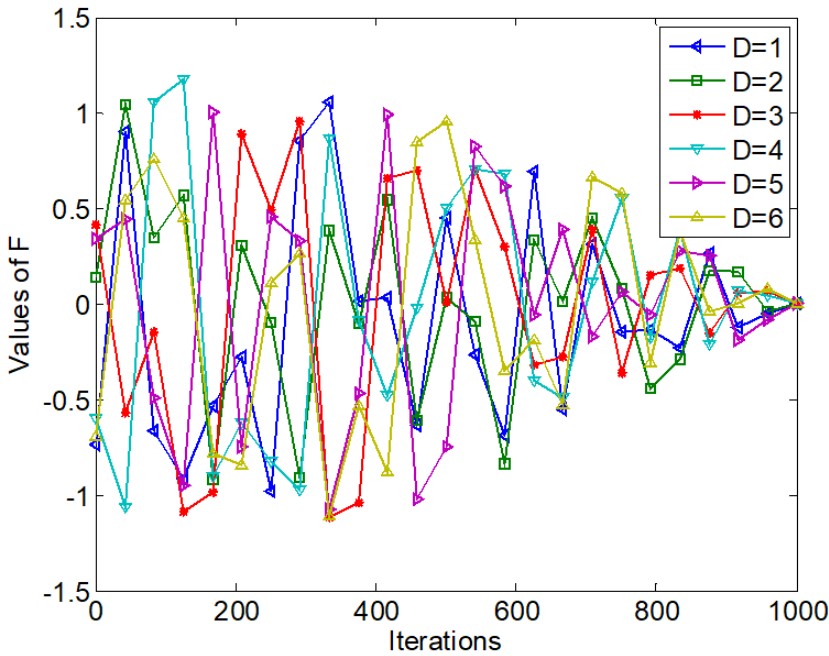

**Figure 2** Values of $F$ with $t$.

$D = 3$. The value of the 3-rd dimension of $F$ changes both randomly and dynamically, and it fluctuates in a large range in the early stage; then it gradually shrinks to 0 as the iteration number increases. Correspondingly, 1- $F$ fluctuates between $[-0.5, 2.5]$ and eventually shrinks to 1 with the iteration increase.

Generation rate $G$ can be calculated as:

$$G = G_0 \otimes F \tag{5}$$

$$G_0 = R_G \times (C_e - \lambda \otimes C_i) \tag{6}$$

$$R_G = \begin{cases} 0.5 r_1 & r_2 \geq P_G \\ 0 & r_2 < P_G \end{cases} \tag{7}$$

where $R_G$ is the generation rate control parameter and is determined by $P_G$. $P_G$ is Generation Probability and a parameter. $r_1$ and $r_2$ are random numbers in $[0, 1]$. From Eqs. (5), (6) and (7), $G$ is a zero vector when $r_2$ is less than $P_G$.

From Eq. (1), the first term on the right of the equation is the concentration of the elite particle, and the second term is the difference between the elite particle's and the $i$-th particle's concentration. The third term is the generation item. Its pseudo-code is shown in Algorithm1 EO. From the above description, EO has the following good features. (1) Unique search mechanism: Each particle's concentration is updated in a unique updating way including the elite particle selected randomly from an elite group. (2) High degree of information guidance: The three items on the right of Eq. (1) all contain elites guiding each particle. (3) Strong local search ability: The guidance of the elite particles makes EO evolve

in a good direction throughout the iteration process. Each component value of $F$ is small in the late iteration stage. The difference between the elite particle and other particles is small, and the value of $G$ is small. Meanwhile, the second term on the right side of Eq. (1) takes up less weight. Moreover, the particle is closer to the elite particle when $R_G$ is 0. These help EO search more accurately. (4) Global search ability. Each component value of $F$ and the difference between the elite particles and other particles are large in the early search stage. Each component value of $G$ is also large when $\lambda$ randomly takes a small value and $R_G$ is not 0. At this time, the value of the third term is also larger, which helps the current particles search the entire search space.

However, there are some shortcomings when EO is used to solve complex OPs. (1) From Eq. (1), the particles in EO learn from the elite particles, and the generation term contributes to the accurate search. This will make EO easy to fall into local optima and have poor exploration ability. (2) From Eq. (1), the particles in the population only communicate with elite particles. This results in a low degree of information sharing and poor diversity in the population. Moreover, EO only learns from elite particles and is also monotonous, which has no other way of guiding and is not conducive to the further evolution of the population. It leads to insufficient search ability. (3) The computational load of the exponential function is higher. Moreover, $\lambda$ is a random vector. It calculates the exponential term $F$ more complicated. (4) EO has many parameters, such as $\lambda$, $F$, $r_1$, $r_2$, $r$, $V$, $a_1$, $a_2$ and $P_G$. Although the parameters increase the flexibility of EO, it leads to poor operability of EO.

## MATERIALS & METHODS

As mentioned in the previous section, EO has a number of advantages when it comes to solving classic OPs (*Faramarzi et al., 2020*). However, there are still drawbacks when solving complex OPs. Therefore, this article proposes a Multi-strategy Synthetized EO (MS-EO) to cope with the above problems of EO.

### Simplified updating strategy

In order to reduce the computational complexity and improve the operability of EO, a simplified updating strategy is used in EO to form a simplified EO (SEO). Its expression is as follows:

$$C_i = C_e + (C_i - C_e) \otimes F + 0.5 \times r_1 \times (C_e - C_i) \otimes F/(1-F) \tag{8}$$

$$F = 2 \times sign(r - 0.5) \times [e^{-\varepsilon} - 1] \tag{9}$$

$$\varepsilon = (1 - t/T)^{(t/T)}. \tag{10}$$

Figure 3 shows the values of simplified $F$.
From Eqs. (8), (9) and (10) and compared with EO, the simplified updating strategy has the following differences:

(1) The simplified updating strategy removes the parameters $\lambda$. Each component value of $F$ in Fig. 3 is more regular and the curves are smoother compared with Fig. 2. Therefore, the simplified updating strategy reduces the computational complexity of EO.

**Peer**J Computer Science

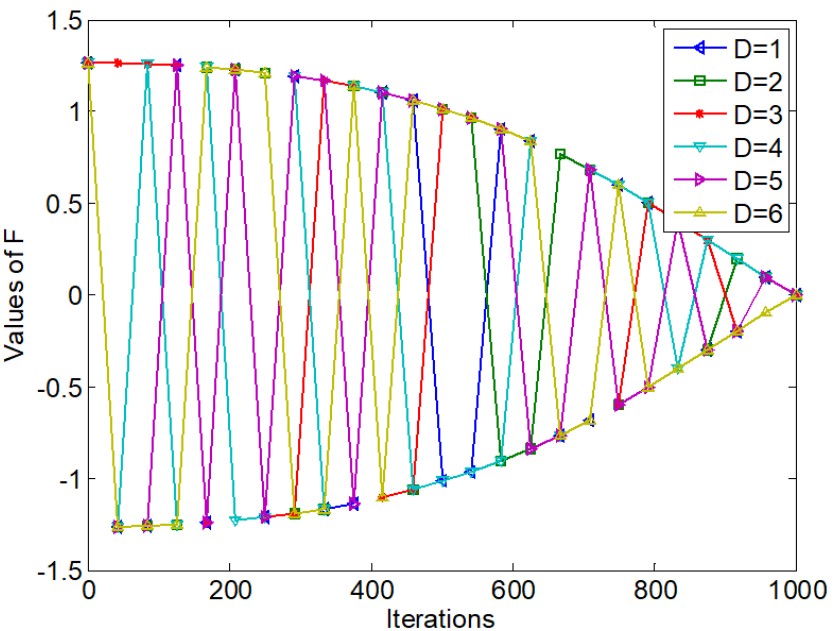

**Figure 3** Values of simplified *F* with *t*.

(2) The generation term in Eq. (8) does not have a value of 0 compared with Eq. (1). The computational load descends to a certain extent.

(3) From Eq. (8), the simplified updating equation highlights the elite particles even more and also highlights the exploitation.

In summary, the simplified updating strategy is the modification of the updating equation. It can effectively reduce the computational complexity of EO and improve its operability. However, the simplified updating strategy does not change the overall search pattern of EO. Therefore, it still has insufficient search ability.

## Information sharing strategy

Whether the information of individuals in the population is fully utilized or not is crucial to the search ability of a population-based MA. The performance of the MA will be significantly improved if the information can be fully utilized in the search process (*Zhang & Yang, 2021*). In order to enhance the information sharing and the global search ability of EO, this article introduces an information sharing strategy. The mathematical model of the information sharing strategy is as follows:

$$\eta = C_a - C_b \tag{11}$$

$$C_i = C_i + f_r \times \eta \tag{12}$$

$$f_r = 0.5 \times (\sin(2\pi \times 0.25 \times t) \times t/T + 1) \tag{13}$$

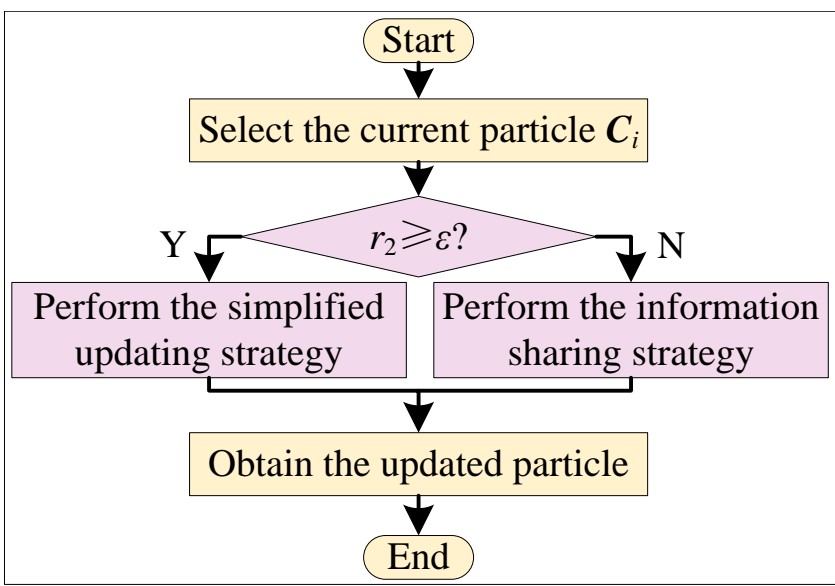

**Figure 4** Flowchart of SS-EO.

where $C_a$ and $C_b$ are the concentration of two particles selected randomly from the current population, $i \neq a \neq b$ and inspired by *Zhang et al. (2020b)*, the scale factor $f_r$ adopts the dynamically adjusted sinusoidal approach.

From Eqs. (11) and (12), a new solution is generated based on the combined action of three different solutions. $\eta$ is the difference between two random solutions, which realizes the information sharing among particles in the population, because all the particles in the population may be selected with the increase of the iteration number $t$, it makes full use of the information of the whole population and helps increase the population diversity.

The simplified updating strategy still maintains the strong exploitation ability of EO. In order to enhance the global search ability of EO, the information sharing strategy and the simplified updating strategy are combined by a dynamical adjustment strategy to form a simplified sharing EO (SS-EO). The flow chart is shown in Fig. 4. According to Eq. (10), $\varepsilon$ is a dynamically adjusted parameter of the simplified strategy that linearly decreases from 1 to 0.

From Fig. 4, the current particle is updated with the simplified updating equation when $r_2$ is greater than or equal to $\varepsilon$. Otherwise, the current particle uses the information sharing updating equation to update. In the early iteration stage, the value of $\varepsilon$ is larger. Most of the particles are updated by the information sharing strategy, and a few of them are updated in a simplified way. The value of $\varepsilon$ is small in the late iteration stage. Most of the particles are updated by the simplified updating way with strong exploitation ability, and a small number of particles are updated by the information sharing strategy.

## Golden particle migration strategy

The golden section is a proportional relationship in mathematics. It enables us to reasonably arrange fewer test times. The migration operator in BBO (*Zhang et al., 2019*) is to share the characteristics of good individuals with poor individuals. It can effectively improve the search ability of poor individuals, so as to enhance the search ability of the whole population. Inspired by the above and in order to stimulate and enhance the search ability, a golden particle migration strategy is introduced SS-EO to form a golden SS-EO (GS-EO). The mathematical model of the migration strategy is as follows:

$$C_g^j = C_q^j \tag{14}$$
$$g = ceil(0.618 \times N) \tag{15}$$
$$q = ceil((g-1) \times rand) \tag{16}$$

where $C$ represents the $j$-th value of the concentration of the golden particle, $C$ is the $j$-th value of the example particle, $q \in [1, g\text{-}1]$, and $j \in [1, D]$. $g$ is the index of the sorted particles from the best to the worst according to the fitness values of all the particles. *ceil* is round-up function. The illustration of the golden particle migration strategy is shown in Fig. 5.

From Eqs. (14), (15) and (16), an example pool is composed of the particles that are better than the golden particle. From Fig. 5, each dimension value of the golden particle comes from the corresponding dimension value of the example particle selected randomly from the example pool. The golden particle migration strategy is individual-based addition. The golden particle requires $D$ example particles. At the same time, each dimension of the golden particle is derived from its example particle also makes it evolve in a good direction. The golden particle gradually becomes excellent as the iteration number increases. And it may be selected as the learning object during the iterative process by the simplified updating strategy and information sharing strategy to participate in the updating for other particles. This further stimulates the search for other particles, so as to improve the search ability of the algorithm.

## Elite learning strategy

Educators often raise the grades of poor students by letting excellent students help poor students, so as to improve the overall grades of the class. Inspired by this and in order to enhance the local search ability, an elite learning strategy is used for the worst particle in the late iteration stage to form MS-EO. The mathematical expression of the elite learning strategy is as follows:

$$\theta = C_e - C_w \tag{17}$$
$$C_w = C_w + f_r \times \theta \tag{18}$$

where $C_w$ is the concentration of the worst particle in the current population.

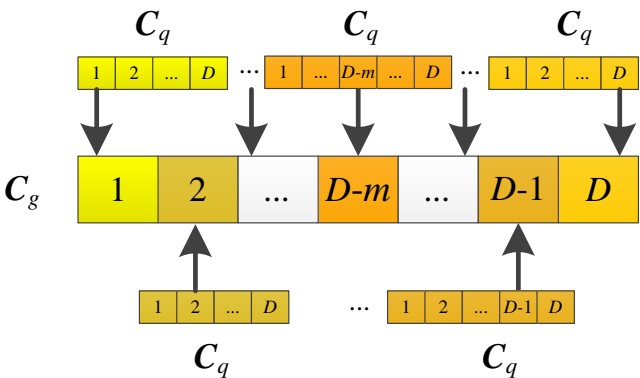

**Figure 5** Illustration of the golden particle migration strategy.

From Eqs. (17) and (18), the worst particle learns from the elite particle to make up for its shortcomings. And it is used in the late stage. Therefore the elite learning strategy belongs to individual-based addition and iteration-based addition. In the late iteration stage, the component value of $\theta$ is smaller and the value of $f_r$ fluctuates widely. When the value of $f_r$ is small, it is helpful to the local search of the algorithm. It can avoid the algorithm falling into local optima when the value of $f_r$ is large. Moreover, the worst particles learn from the elite particles. It greatly enhances the exploitation ability of the worst particle, so as to enhance the search ability of the whole population.

## Framework of the proposed algorithm

MS-EO skillfully synthesizes many strategies, such as simplifying, information sharing, golden particle migration and elite learning, and embeds them into EO according to their respective characteristics. The flow chart of MS-EO is shown in Fig. 6.

From Fig. 6, the information sharing strategy is mainly used to update the most particles' concentrations in the early iteration stage. In the late iteration stage, most particles are mainly updated in a simplified way. For the golden particle, the migration strategy is adopted to make it originate from the example particles. For the worst particle, the elite learning strategy is adopted in the late iteration stage to further improve the optimization level of the whole population.

## Complexity analysis

On time complexity, it is assumed that the updating equation of EO requires $L_0$ time unit to generate a variable of a new solution, while the updating equations of the simplified updating strategy, information sharing strategy, golden particle migration strategy and elite learning strategy require $L_1$, $L_2$, $L_3$, and $L_4$ to generate a variable of a new solution, respectively. According to the described above, the simplified updating strategy removes some components in EO. Eq. (8) has fewer operations compared with Eq. (1). Thus, Eq. (8) takes much less time to generate a new solution than Eq. (1). The updating equations of the remaining three strategies are all simpler than Eq. (1). Therefore, $L_1$, $L_2$, $L_3$, and $L_4$ are much less than $L_0$. MS-EO has lower time complexity.

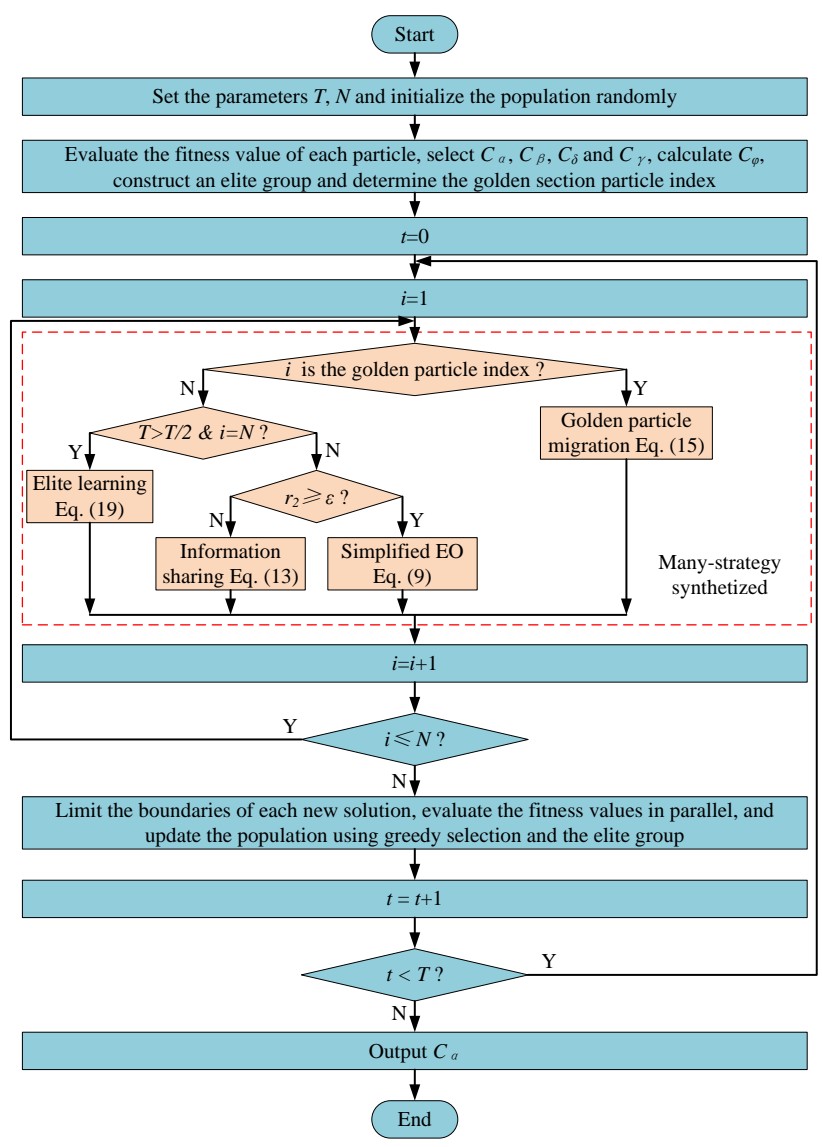

**Figure 6 Flowchart of MS-EO.**

For space complexity, the space to be allocated for EO includes current population, elite pool and fitness value. MS-EO does not allocate additional storage space. Its space complexity is the same as that of EO.

## RESULTS

### Experimental environment and evaluation standard

In order to verify MS-EO, a large number of experiments are performed on the complex functions from CEC2013 and CEC2017 test sets. Their details can be found in *Liang et al. (2012)*; *Awad et al. (2016)*. CEC2013 test set includes 28 functions, among which $f_1 \sim f_5$ are unimodal functions, $f_6 \sim f_{20}$ are basic multimodal functions, and $f_{21} \sim f_{28}$ are composition

functions. CEC2017 test set contains four types of functions: unimodal functions ($f_1 \sim f_3$), simple multimodal functions ($f_4 \sim f_{10}$), hybrid functions ($f_{11} \sim f_{20}$) and composition functions ($f_{21} \sim f_{30}$). Compared with CEC2013, CEC2017 is more complex, such as more function types and more shifted functions. All experiments were conducted in MATLAB (R2014a) environment running on a PC with 3.4 GHz i7-3770 CPU and 8GB RAM. The statistical analysis software was IBM SPSS Statistics 19.

This article adopts statistics such as mean value (Mean), standard deviation (Std), ranking (Rank), and average ranking (Ave. Rank) to evaluate the optimization performance of an algorithm (*Zhang et al., 2019*). The algorithm with a smaller Mean value has better performance for a minimum problem in this article. The ranking rule is as follows, On each function, the less the Mean value obtained by the algorithm is, the higher the ranking is; if the Mean values by the algorithms are the same, the higher the ranking is and the less the Std value is; if the Mean and Std values of the algorithms are the same, then the algorithms rank the same. The best data in the table is bold.

## Comparison algorithms and the parameter settings

There is less literature on improvement on EO since it is a newly proposed algorithm. Therefore, some state-of-the-art algorithms are selected as the comparison algorithms. These algorithms include TPC-GA (*Elsayed, Sarker & Essam, 2013*), YY-FA (*Wang et al., 2020*), DS-PSO (*Zhang et al., 2019*), DC-FWA (*Wei et al., 2021*), DPC-ABC (*Cui et al., 2018*), FSS-DA (*Han & Zhu, 2020*), BHCS (*Chen & Yu, 2019*), AMop-GA (*Lim, Al-Betar & Khader, 2015*), DQL-SFLA (*Zhang et al., 2019*), Sa-DE (*Qin, Huang & Suganthan, 2009*), ME-GWO (*Tu, Chen & Liu, 2019*), A-EO (*Wunnava et al., 2020*), HFPSO (*Aydilek, 2018*), HBBOG (*Zhang et al., 2018*), MSS-CS (*Long et al., 2018*). Their simple description is shown in Table 1. They not only include some improved algorithms of classic algorithms GA, PSO, DE and FWA, but also some improved variants of the latest algorithms.

For a fair comparison, the common parameter settings of all the algorithms are the same, such as the maximum number of function evaluations ($M_{nfe}$), the number of independent runs ($N_{run}$), and *D*. According to the best recommendation of (*Liang et al., 2012*; *Awad et al., 2016*), $M_{nfe}$ is set to $D \times 10000$ and $N_{run}$ is 51. Different parameter settings of an algorithm will result in different experimental results, so other parameter settings of the comparison algorithms are set by the best settings from the corresponding references shown in Table 1. In addition, all algorithms assume that the experiment is carried out in an environment without interference from other factors.

## Comparison with its incomplete algorithms

In order to demonstrate the contribution of each new strategy of MS-EO, this subsection compares MS-EO with EO, SS-EO and GS-EO on the 30-dimensional functions from CEC2013 test set. In order to explain the problem briefly, this subsection selects the experimental results of three representative functions to illustrate. The three representative functions are unimodal function ($f_3$), basic multimodal function ($f_7$) and composition function ($f_{22}$). Fig. 7 provides a performance graph of MS-EO and incomplete variants on the 3 representative functions and an average ranking chart on all the functions,

**Table 1  Specific parameter settings of the algorithms.**

| No. | Algorithm | Ref. | Year | Simple description | Parameter setting |
|---|---|---|---|---|---|
| 01 | MS-EO | | 2021 | The proposed algorithm | CEC2013: $N = 100$; CEC2017: $N = 80$. |
| 02 | DC-FWA | [30] | 2021 | Dynamic collaborative fireworks algorithm | $Ca = 1$, $Cr = 0.9$, $a = r = 0.1$. |
| 03 | MSS-CS | [15] | 2021 | Multi-strategy serial CS | $N = 25$, $\alpha = 0.01$, $\beta = 1.5$, $P_a = 0.25$, $c = 0.2$, $PA_{max} = 0.35$, $PA_{min} = 0.25$. |
| 04 | YY-FA | [29] | 2020 | Yin-yang firefly algorithm with Cauchy mutation | $N = 30$, $L = 800$, $\beta_0 = 1$; $\beta_{min} = 0.2$; $\alpha(0) = 0.2$; $\gamma = 1$. |
| 05 | FSS-DA | [32] | 2020 | Fusion with distance-aware selection strategy for dandelion algorithm | $N = 5$, $M_S = 5$, $r = 0.95$, $e = 1.05$. |
| 06 | A-EO | [18] | 2020 | Adaptive EO | $N = 100$, $V = 1$, $a_1 = 2$, $a_2 = 1$, $G_P = 0.5$. |
| 07 | DS-PSO | [16] | 2019 | PSO with single and mean example learning strategies | $N = 40$, $c_1 = 2$, $F_{min} = 0.7$. |
| 08 | DQL-SFLA | [35] | 2019 | Lévyflight SFLA with differential perturbation and quasi-Newton search | $N = 20$, $m = 5$. |
| 09 | BHCS | [33] | 2019 | Hybrid CS with BBO | $N = 20$, $pa = 0.3$, $\alpha = 1.1$, $\beta = 1.7$, $\delta = 1.6$, $I = E = 1$. |
| 10 | ME-GWO | [37] | 2019 | Multi-strategy ensemble GWO | $N = 100$, $GR = 0.8$, $SR_{max} = 1$, $SR_{min} = 0.6$, $DR_{max} = 0.4$, $DR_{min} = 0$. |
| 11 | EO | [14] | 2019 | Equilibrium optimizer | $N = 100$, $V = 1$, $a_1 = 2$, $a_2 = 1$, $G_P = 0.5$. |
| 12 | DPC-ABC | [31] | 2018 | Enhanced ABC with dual-population framework | $N = 50$, $limit = N \times D$, $m = 3.5$, $minSN_{CP} = 20$. |
| 13 | HFPSO | [38] | 2018 | Hybrid firefly algorithm with PSO | $N = 100$, $a = 0.2$, $B_0 = 2$, $\gamma = 1$, $c_1 = c_2 = 1.49445$, $\omega_i = 0.9$, $\omega_f = 0.5$. |
| 14 | HBBOG | [39] | 2018 | Hybrid BBO with GWO | $N = 100$, $I = 1$. |
| 15 | AMop-GA | [34] | 2015 | Adaptive monogamous pairs GA | $N = 10$, $\Phi = 0.01$, $\alpha = 0.5$. |
| 16 | TPC-GA | [28] | 2015 | GA with three parents crossing | $N = 90$, $\gamma = 40$, $\xi = 1.0E-08$, $PS_{add} = N$, $\Omega = 3 \times PS_{add}$. |
| 17 | Sa-DE | [36] | 2009 | Self-adaptive DE | $N = 20$, $F\text{-}N \in (0.5, 0.3)$, $CR\text{-}N \in (CR_m, 0.1)$. |

respectively. The parameters settings for MS-EO and the incomplete variants are set to the same.

From Fig. 7D, the corresponding ranking order is MS-EO, GS-EO, SS-EO and EO on all the functions. From Figs. 7A, 7B and 7C, the information sharing strategy, the simplified update strategy and the gold particle migration strategy all improve the performance of EO to a certain extent, and the adoption of the elite learning strategy for the worst particles in the late iteration stage can effectively enhance the local search ability of SS-EO, which indicates that the optimization performance of MS-EO is better than that of GS-EO, especially on $f_7$. It shows that every improvement of MS-EO is effective.

## Performance comparison on CEC2013 test set

In order to verify the ability of MS-EO to solve the complex problems, this subsection compares the results of MS-EO with those of EO and some state-of-the-art comparison algorithms on CEC2013 test set. The comparison algorithms include TPC-GA, YY-FA, DS-PSO, DC-FWA, DPC-ABC, FSS-DA, BHCS and AMop-GA. To illustrate intuitively,

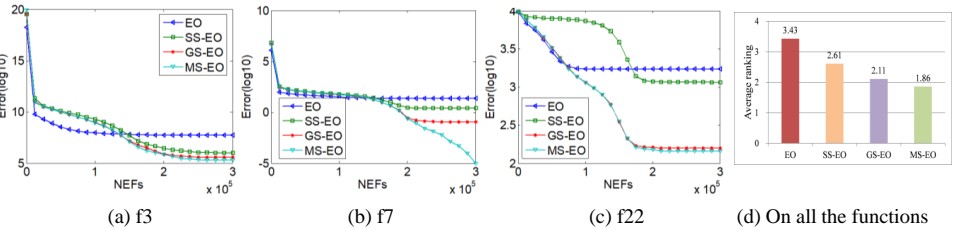

(a) f3       (b) f7       (c) f22       (d) On all the functions

**Figure 7** Performance curves and average ranking chart of MS-EO with EO and its incomplete variants.

Fig. 8 provides the stacked histogram of ranking for MS-EO and 9 comparison algorithms on the 30-dimensional functions from CEC2013 test set. Table 2 lists their Friedman test results. The experimental results of EO and MS-EO are from our experiments, while the results of the other 8 algorithms are taken from *Elsayed, Sarker & Essam (2013)*, *Wang et al. (2020)*, *Zhang et al. (2019)*, *Wei et al. (2021)*, *Cui et al. (2018)*, *Han & Zhu (2020)*, *Zhang et al. (2020a)*, *Zhang et al. (2020b)* and *Lim, Al-Betar & Khader (2015)*.

In Fig. 8, the height of each color block represents the times obtained by an algorithm in ranking the $x$-th. For example, '①' represents ranking the first. The higher the height of the color block is the more times obtained by the algorithm in ranking first. It is obvious that these three color blocks ('①', '②' and '③') have the highest height in MS-EO. Therefore, MS-EO has the best optimization performance. From Table 2, on Ave. Rank, the value of EO is 6.04, ranking the eighth among all the 10 algorithms. MS-EO ranks first with Ave. Rank of 3.29. It indicates that the improvements proposed in this article greatly enhance the optimization performance of EO.

## Performance comparison on the CEC2017 test set

In order to further verify the ability of MS-EO to solve more complex problems, many experiments are conducted on the more complex functions from the CEC2017 test set. Some representative algorithms are selected for comparison. These algorithms include EO, DQL-SFLA, Sa-DE, ME-GWO, A-EO, HFPSO, HBBOG, DS-PSO and MSS-CS. The parameters of the algorithms are shown in Table 1. Tables 3 and 4 provide the comparison results of MS-EO and these representative algorithms on the 30- and 50-dimensional functions from CEC2017 test set, respectively. Among them, the data of Sa-DE is directly taken from *Tang (2019)*.

From Table 3, on Ave. Rank, EO is 7.33 and ranks ninth. It is better than HFPSO (8.60). On unimodal functions, MSS-CS ranks first on $f_1$, MS-EO ranks the first on $f_2$ and $f_3$, and other comparison algorithms obtain no times ranking the first. It means that MS-EO has strong exploitation ability. DS-PSO obtains four times ranking first on the multimodal functions. It shows the strong exploration ability of DS-PSO. However, MS-EO can obtain one time ranking first and four times ranking second on the multimodal functions. It indicates that MS-EO also has strong exploration ability compared with the other comparison algorithms. MS-EO won six times and three times ranking first on the hybrid and composition functions, respectively. However, from the overall ranking,

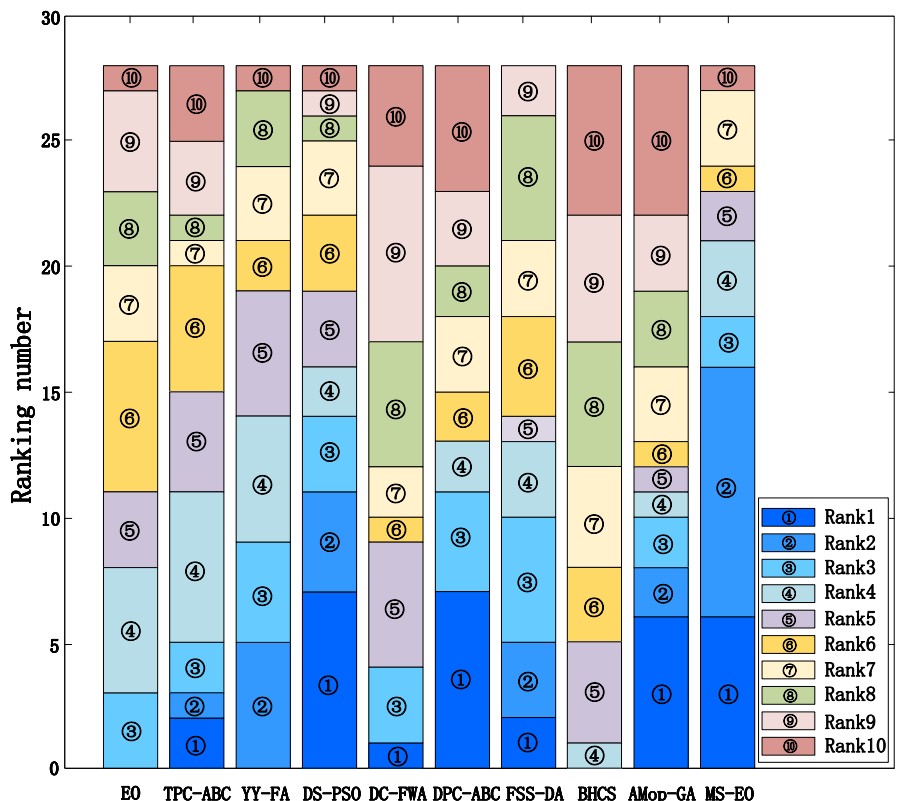

**Figure 8** Stacked histogram of ranking for MS-EO and the comparison algorithms on the CEC2013 test set ($D = 30$).

**Table 2** Friedman test results on the CEC2013 test set ($D = 30$).

| Algorithm | Ave. rank | Rank | *p*-value |
|---|---|---|---|
| EO | 6.04 | 8 | |
| TPC-GA | 5.61 | 6 | |
| YY-FA | 4.79 | 3 | |
| DS-PSO | 4.04 | 2 | |
| DC-FWA | 7.09 | 9 | 4.6868E−07 |
| DPC-ABC | 5.57 | 5 | |
| FSS-DA | 5.09 | 4 | |
| BHCS | 7.63 | 10 | |
| Amop-GA | 5.88 | 7 | |
| MS-EO | **3.29** | **1** | |

MS-EO obtains 12 times ranking the first, and eight times ranking the second. Moreover, on Ave. Rank, the value of MS-EO is 2.77. Therefore, compared with these comparison algorithms, MS-EO has strong optimization ability on the 30-dimensional functions from the CEC2017 test set.

**Table 3** Comparison results of MS-EO and nine comparison algorithms on the CEC2017 test set ($D = 30$).

| Fun. | Metric | EO | DQL-SFLA | Sa-DE | ME-GWO | A-EO | HFPSO | HBBOG | DS-PSO | MSS-CS | MS-EO |
|---|---|---|---|---|---|---|---|---|---|---|---|
| $f_1$ | Mean | 3.3222E−03 | 7.2935E−01 | 3.0714E−03 | 4.5517E−03 | 4.8353E−03 | 3.9338E−03 | 2.2211E−00 | 1.6209E−03 | **1.4768E−14** | 6.3447E−13 |
| | Std | 4.3491E−03 | 1.8488E−02 | 3.5072E−03 | 1.0677E−03 | 5.8456E−03 | 5.3689E−03 | 2.5455E−00 | 1.9165E−03 | **2.7859E−15** | 1.3821E−12 |
| | Rank | 7 | 4 | 6 | 9 | 10 | 8 | 3 | 5 | **1** | 2 |
| $f_2$ | Mean | 1.9754E−09 | 9.9655E−13 | 8.6275E−01 | 2.8884E−08 | 3.3117E−13 | 5.3485E−04 | 3.0644E−05 | 4.2568E−08 | 4.1898E−05 | **4.3161E−09** |
| | Std | 1.2191E−10 | 4.0151E−14 | 4.9357E−00 | 8.0571E−08 | 9.6854E−13 | 3.2847E−05 | 1.7484E−06 | 3.0034E−09 | 1.6051E−06 | **1.2616E−08** |
| | Rank | 8 | 10 | 2 | 6 | 9 | 3 | 4 | 7 | 5 | **1** |
| $f_3$ | Mean | 2.3527E−01 | 2.3044E−05 | 3.0045E−02 | 2.2633E−02 | 1.2905E−03 | 1.5595E−07 | 2.4721E−01 | 8.0916E−06 | 1.3736E−02 | **2.8310E−13** |
| | Std | 6.7349E−01 | 2.1837E−05 | 7.3017E−02 | 1.7031E−02 | 1.1722E−03 | 2.3334E−07 | 2.7490E−01 | 1.8234E−05 | 1.3680E−02 | **1.9642E−13** |
| | Rank | 5 | 4 | 9 | 8 | 10 | 2 | 6 | 3 | 7 | **1** |
| $f_4$ | Mean | 6.9612E−01 | **1.5634E−00** | 6.0423E−01 | 2.4815E−01 | 9.1893E−01 | 6.9386E−01 | 1.2058E−01 | 1.1973E−02 | 1.7826E−01 | 4.8313E−01 |
| | Std | 2.9970E−01 | **1.9658E−00** | 2.9825E−01 | 2.8995E−01 | 5.5709E−01 | 2.1364E−01 | 2.3310E−01 | 1.0171E−01 | 2.7170E−01 | 2.8941E−01 |
| | Rank | 8 | **1** | 6 | 4 | 9 | 7 | 2 | 10 | 3 | 5 |
| $f_5$ | Mean | 5.7667E−01 | 6.8633E−01 | 5.6192E−01 | 5.6912E−01 | 1.4209E−02 | 8.5624E−01 | 4.8262E−01 | **1.6797E−01** | 5.4730E−01 | 3.6135E−01 |
| | Std | 1.6196E−01 | 1.9409E−01 | 1.4216E−01 | 1.0725E−01 | 3.3861E−01 | 1.7427E−01 | 7.8631E−00 | **4.7117E−00** | 1.1402E−01 | 8.8507E−00 |
| | Rank | 7 | 8 | 5 | 6 | 10 | 9 | 3 | **1** | 4 | 2 |
| $f_6$ | Mean | 5.6559E−02 | 1.5171E−01 | 8.9317E−02 | 2.4470E−01 | 4.0061E−02 | 1.0170E−00 | **1.6161E−06** | 2.1338E−03 | 8.9878E−04 | 8.2332E−06 |
| | Std | 1.2024E−01 | 3.3472E−00 | 1.3955E−01 | 8.1620E−02 | 5.0597E−02 | 2.3644E−00 | **2.2666E−06** | 1.0733E−02 | 1.7816E−03 | 1.1509E−05 |
| | Rank | 6 | 10 | 7 | 8 | 5 | 9 | **1** | 4 | 3 | 2 |
| $f_7$ | Mean | 8.8332E−01 | 1.2197E−02 | 9.4945E−01 | 8.9106E−01 | 8.9058E−01 | 1.0407E−02 | 7.8879E−01 | **4.0294E−01** | 9.1156E−01 | 6.4465E−01 |
| | Std | 1.8194E−01 | 2.6617E−01 | 1.9879E−01 | 1.0935E−01 | 1.7893E−01 | 1.9791E−01 | 6.1638E−00 | **2.9776E−00** | 9.2863E−00 | 1.4880E−01 |
| | Rank | 4 | 10 | 8 | 6 | 5 | 9 | 3 | **1** | 7 | 2 |
| $f_8$ | Mean | 5.7110E−01 | 6.6428E−01 | 5.3942E−01 | 5.9398E−01 | 5.3384E−01 | 7.2842E−01 | 5.4157E−01 | **1.5997E−01** | 5.8868E−01 | 3.5646E−01 |
| | Std | 1.3610E−01 | 1.4052E−01 | 1.2792E−01 | 1.0663E−01 | 1.3469E−01 | 1.7967E−01 | 8.0374E−00 | **4.3271E−00** | 8.8901E−00 | 1.1128E−01 |
| | Rank | 6 | 9 | 4 | 8 | 3 | 10 | 5 | **1** | 7 | 2 |
| $f_9$ | Mean | 3.2070E−01 | 5.5931E−02 | 8.3556E−01 | 7.8267E−00 | 5.9311E−00 | 3.2733E−01 | 4.4586E−01 | 6.0472E−02 | 6.7240E−00 | **1.1815E−13** |
| | Std | 8.1996E−01 | 1.9193E−02 | 6.2643E−01 | 1.1815E−01 | 1.7346E−01 | 1.3249E−02 | 4.2698E−01 | 1.7766E−01 | 6.9160E−00 | **2.2287E−14** |
| | Rank | 7 | 10 | 9 | 6 | 4 | 8 | 3 | 2 | 5 | **1** |
| $f_{10}$ | Mean | 3.1597E−03 | 3.1971E−03 | 2.3253E−03 | 2.4369E−03 | 3.2479E−03 | 2.9908E−03 | 1.9872E−03 | **1.8489E−03** | 2.2401E−03 | 4.0220E−03 |
| | Std | 5.9970E−02 | 5.4089E−02 | 4.9247E−02 | 4.4542E−02 | 7.4246E−02 | 5.9210E−02 | 3.0062E−02 | **5.4082E−02** | 2.8344E−02 | 7.3440E−02 |
| | Rank | 7 | 8 | 4 | 5 | 9 | 6 | 2 | **1** | 3 | 10 |
| $f_{11}$ | Mean | 5.9497E−01 | 4.3137E−01 | 1.0032E−02 | 2.9612E−01 | 6.0588E−01 | 1.1553E−02 | 3.8609E−01 | 8.2135E−01 | 1.9435E−01 | **1.7248E−01** |
| | Std | 4.2844E−01 | 1.5616E−01 | 4.3101E−01 | 1.0347E−01 | 3.1569E−01 | 3.9628E−01 | 2.6323E−01 | 2.9775E−01 | 6.6436E−00 | **7.1973E−00** |
| | Rank | 6 | 5 | 9 | 3 | 7 | 10 | 4 | 8 | 2 | **1** |

*(continued on next page)*

Sun et al. (2024), *PeerJ Comput. Sci.*, DOI 10.7717/peerj-cs.1760

**Table 3** (*continued*)

| Fun. | Metric | EO | DQL-SFLA | Sa-DE | ME-GWO | A-EO | HFPSO | HBBOG | DS-PSO | MSS-CS | MS-EO |
|---|---|---|---|---|---|---|---|---|---|---|---|
| | Mean | 6.6389E−04 | 2.0443E−03 | 6.8629E−04 | 1.5983E−04 | 3.5529E−05 | 9.9670E−04 | 3.8672E−04 | 1.9817E−04 | 4.7056E−04 | **2.8776E−02** |
| $f_{12}$ | Std | 4.3631E−04 | 8.4234E−02 | 3.8252E−04 | 4.0434E−03 | 4.0698E−05 | 1.0658E−05 | 2.3412E−04 | 1.1542E−04 | 2.0525E−04 | **1.5623E−02** |
| | Rank | 7 | 2 | 8 | 3 | 10 | 9 | 5 | 4 | 6 | **1** |
| | Mean | 2.1679E−04 | 8.9030E−02 | 1.1211E−04 | 2.0450E−02 | 2.3817E−04 | 3.0927E−04 | 3.6491E−03 | 8.3675E−03 | 6.9328E−01 | **3.2490E−01** |
| $f_{13}$ | Std | 1.6448E−04 | 2.5263E−02 | 1.0535E−04 | 2.7028E−01 | 2.0917E−04 | 2.7301E−04 | 3.8431E−03 | 7.4967E−03 | 1.9798E−01 | **1.3414E−01** |
| | Rank | 8 | 4 | 7 | 3 | 9 | 10 | 5 | 6 | 2 | **1** |
| | Mean | 5.6825E−03 | 4.4698E−01 | 4.3238E−03 | 6.1985E−01 | 2.0635E−04 | 6.7377E−03 | 9.5505E−01 | 7.8420E−02 | 4.3572E−01 | **3.5387E−01** |
| $f_{14}$ | Std | 5.0922E−03 | 7.4124E−00 | 5.7159E−03 | 8.6647E−00 | 1.9593E−04 | 5.5695E−03 | 2.0063E−01 | 6.7967E−02 | 6.2658E−00 | **4.2706E−00** |
| | Rank | 8 | 3 | 7 | 4 | 10 | 9 | 5 | 6 | 2 | **1** |
| | Mean | 5.5192E−03 | 1.0898E−02 | 2.1676E−03 | 5.1634E−01 | 5.8095E−03 | 9.7487E−03 | 1.1499E−02 | 2.6350E−03 | 2.1055E−01 | **1.3851E−01** |
| $f_{15}$ | Std | 5.9116E−03 | 3.0585E−01 | 3.0178E−03 | 1.0713E−01 | 6.6724E−03 | 1.2114E−04 | 3.1105E−01 | 2.7534E−03 | 6.0683E−00 | **5.3627E−00** |
| | Rank | 8 | 4 | 6 | 3 | 9 | 10 | 5 | 7 | 2 | **1** |
| | Mean | 6.8440E−02 | 4.7544E−02 | 5.6072E−02 | **4.4823E−02** | 4.8567E−02 | 7.7229E−02 | 4.6637E−02 | 4.9749E−02 | 5.0563E−02 | 6.6557E−02 |
| $f_{16}$ | Std | 2.8098E−02 | 2.3772E−02 | 2.0850E−02 | **1.3443E−02** | 2.3847E−02 | 2.2590E−02 | 1.7908E−02 | 1.6456E−02 | 1.5740E−02 | 2.3371E−02 |
| | Rank | 9 | 3 | 7 | **1** | 4 | 10 | 2 | 5 | 6 | 8 |
| | Mean | 2.2150E−02 | 1.5303E−02 | 8.7684E−01 | **6.9544E−01** | 1.3676E−02 | 2.5591E−02 | 8.7145E−01 | 1.5260E−02 | 8.9996E−01 | 1.2949E−02 |
| $f_{17}$ | Std | 1.7044E−02 | 6.4439E−01 | 9.1289E−01 | **1.7296E−01** | 7.1383E−01 | 1.2971E−02 | 5.0306E−01 | 5.4151E−01 | 4.7714E−01 | 7.9698E−01 |
| | Rank | 9 | 8 | 3 | **1** | 6 | 10 | 2 | 7 | 4 | 5 |
| | Mean | 1.2375E−05 | 6.7577E−01 | 1.0034E−05 | 2.0505E−02 | 2.6997E−05 | 1.1409E−05 | 8.0963E−03 | 4.4368E−04 | 2.9366E−02 | **2.4552E−01** |
| $f_{18}$ | Std | 8.3515E−04 | 1.6128E−01 | 1.1019E−05 | 4.7536E−01 | 2.0513E−05 | 1.1535E−05 | 6.3692E−03 | 2.1042E−04 | 3.8250E−02 | **1.4356E−00** |
| | Rank | 9 | 2 | 7 | 3 | 10 | 8 | 5 | 6 | 4 | **1** |
| | Mean | 7.5536E−03 | 2.4696E−01 | 5.9612E−03 | 2.9977E−01 | 6.6152E−03 | 8.6631E−03 | 6.4050E−01 | 4.3373E−03 | **1.4446E−01** | 1.7041E−01 |
| $f_{19}$ | Std | 9.2361E−03 | 4.1265E−00 | 7.1112E−03 | 3.3897E−00 | 1.0646E−04 | 1.9974E−04 | 2.0384E−01 | 3.5805E−03 | **2.3613E−00** | 2.5649E−00 |
| | Rank | 9 | 3 | 7 | 4 | 8 | 10 | 5 | 6 | **1** | 2 |
| | Mean | 2.0954E−02 | 2.2062E−02 | 1.2989E−02 | **1.1363E−02** | 1.6996E−02 | 2.6516E−02 | 1.5080E−02 | 1.7999E−02 | 1.6235E−02 | 1.6839E−02 |
| $f_{20}$ | Std | 1.2606E−02 | 6.2726E−01 | 7.0970E−01 | **5.2411E−01** | 1.0179E−02 | 1.1737E−02 | 7.7719E−01 | 3.0863E−01 | 8.3552E−01 | 1.0846E−02 |
| | Rank | 8 | 9 | 2 | **1** | 6 | 10 | 3 | 7 | 4 | 5 |
| | Mean | 2.4861E−02 | 2.5401E−02 | 2.4896E−02 | 2.5458E−02 | 2.4845E−02 | 2.7446E−02 | **2.1555E−02** | 2.2234E−02 | 2.4237E−02 | 2.3357E−02 |
| $f_{21}$ | Std | 1.3100E−01 | 2.5306E−01 | 1.3195E−01 | 3.3247E−01 | 1.3575E−01 | 1.9517E−01 | **6.8417E−01** | 5.3235E−00 | 4.8332E−01 | 1.3000E−01 |
| | Rank | 6 | 8 | 7 | 9 | 5 | 10 | **1** | 2 | 4 | 3 |

**Table 3** (*continued*)

| Fun. | Metric | EO | DQL-SFLA | Sa-DE | ME-GWO | A-EO | HFPSO | HBBOG | DS-PSO | MSS-CS | MS-EO |
|---|---|---|---|---|---|---|---|---|---|---|---|
| | Mean | 1.1194E−03 | 1.0052E−02 | 1.0228E−02 | 1.0022E−02 | 9.3847E−02 | 1.4532E−03 | 1.0010E−02 | **1.0000E−02** | 1.0382E−03 | 1.0000E−02 |
| $f_{22}$ | Std | 1.5994E−03 | 1.2385E−00 | 3.2279E−00 | 4.3917E−02 | 1.4111E−03 | 1.8286E−03 | 4.8158E−01 | **1.9609E−13** | 1.2396E−03 | 5.3353E−13 |
| | Rank | 9 | 5 | 6 | 4 | 7 | 10 | 3 | **1** | 8 | 2 |
| | Mean | 4.0719E−02 | 4.3546E−02 | 4.1472E−02 | 3.8959E−02 | 4.0008E−02 | 4.8447E−02 | **3.4974E−02** | 3.6743E−02 | 4.0526E−02 | 3.8643E−02 |
| $f_{23}$ | Std | 1.8730E−01 | 2.2279E−01 | 1.8742E−01 | 6.8787E−01 | 1.5030E−01 | 4.4709E−01 | **1.1018E−02** | 9.0375E−00 | 1.6627E−01 | 2.4054E−01 |
| | Rank | 7 | 9 | 8 | 4 | 5 | 10 | **1** | 2 | 6 | 3 |
| | Mean | 4.7031E−02 | 4.9961E−02 | 4.8169E−02 | 4.8972E−02 | 4.6423E−02 | 5.6079E−02 | 4.6378E−02 | **4.3362E−02** | 4.8916E−02 | 4.4400E−02 |
| $f_{24}$ | Std | 1.6522E−01 | 2.0523E−01 | 2.0610E−01 | 1.6597E−01 | 1.4334E−01 | 5.7847E−01 | 1.3594E−02 | **4.9550E−00** | 4.4886E−01 | 3.9737E−01 |
| | Rank | 5 | 9 | 6 | 8 | 4 | 10 | 3 | **1** | 7 | 2 |
| | Mean | 3.8769E−02 | **3.8286E−02** | 4.0124E−02 | 3.8374E−02 | 3.8764E−02 | 3.8818E−02 | 3.8649E−02 | 3.8769E−02 | 3.8583E−02 | 3.8656E−02 |
| $f_{25}$ | Std | 7.9822E−00 | **1.4018E−01** | 1.9489E−01 | 1.8246E−01 | 8.2423E−00 | 3.4076E−00 | 1.0050E−00 | 4.8364E−01 | 1.6363E−00 | 7.0457E−01 |
| | Rank | 8 | **1** | 10 | 2 | 6 | 9 | 4 | 7 | 3 | 5 |
| | Mean | 1.5490E−03 | 1.2519E−03 | 1.7344E−03 | **2.5051E−02** | 1.3867E−03 | 1.4922E−03 | 2.7819E−02 | 4.0567E−02 | 7.4010E−02 | 3.1174E−02 |
| $f_{26}$ | Std | 2.5666E−02 | 6.8198E−02 | 7.1347E−02 | **4.1112E−01** | 3.3105E−02 | 9.6940E−02 | 1.9359E−02 | 3.4155E−02 | 6.7559E−02 | 1.1411E−02 |
| | Rank | 9 | 6 | 10 | **1** | 7 | 8 | 2 | 4 | 5 | 3 |
| | Mean | 5.1702E−02 | 5.0569E−02 | 5.4289E−02 | 5.1286E−02 | 5.1096E−02 | 5.3523E−02 | 5.0897E−02 | 5.0926E−02 | 5.0139E−02 | **5.0058E−02** |
| $f_{27}$ | Std | 1.0225E−01 | 2.5923E−01 | 1.7086E−01 | 6.1632E−00 | 8.0838E−00 | 2.1095E−01 | 4.7593E−00 | 8.1065E−00 | 9.4424E−00 | **9.5229E−00** |
| | Rank | 8 | 3 | 10 | 7 | 6 | 9 | 4 | 5 | 2 | **1** |
| | Mean | 3.5720E−02 | 3.2849E−02 | 3.3257E−02 | 3.6492E−02 | 4.0282E−02 | 3.5331E−02 | 3.2843E−02 | 3.9331E−02 | 3.0608E−02 | **3.0224E−02** |
| $f_{28}$ | Std | 5.9081E−01 | 5.0025E−01 | 5.2165E−01 | 3.2477E−01 | 2.6602E−01 | 5.9179E−01 | 4.1817E−01 | 5.4248E−01 | 2.4544E−01 | **1.4512E−01** |
| | Rank | 7 | 4 | 5 | 8 | 10 | 6 | 3 | 9 | 2 | **1** |
| | Mean | 6.3631E−02 | 7.4492E−02 | 5.5826E−02 | 5.4385E−02 | 5.8903E−02 | 6.7006E−02 | 5.1223E−02 | **4.7663E−02** | 5.1924E−02 | 6.4501E−02 |
| $f_{29}$ | Std | 1.6299E−02 | 1.2566E−02 | 1.0040E−02 | 5.4241E−01 | 1.0176E−02 | 1.4475E−02 | 5.8019E−01 | **3.9323E−01** | 6.6026E−01 | 8.5013E−01 |
| | Rank | 7 | 10 | 5 | 4 | 6 | 9 | 2 | **1** | 3 | 8 |
| | Mean | 1.1907E−04 | 4.6351E−03 | 5.0147E−03 | 3.6855E−03 | 1.5591E−04 | 1.8733E−04 | 5.4617E−03 | 3.9418E−03 | 2.6103E−03 | **2.0742E−03** |
| $f_{30}$ | Std | 3.7716E−04 | 1.5177E−03 | 1.9712E−03 | 3.3042E−02 | 3.3443E−04 | 3.4470E−04 | 1.0452E−03 | 1.4015E−03 | 3.1169E−02 | **5.0097E−01** |
| | Rank | 8 | 5 | 6 | 3 | 9 | 10 | 7 | 4 | 2 | **1** |
| Best/2nd best | | 0/0 | 2/2 | 0/2 | 4/1 | 0/0 | 0/1 | 3/6 | 7/3 | 2/7 | 12/8 |
| Ave. rank | | 7.33 | 5.90 | 6.53 | 4.73 | 7.27 | 8.60 | 3.43 | 4.43 | 4.00 | 2.77 |
| Total. rank | | 9 | 6 | 7 | 5 | 8 | 10 | 2 | 4 | 3 | 1 |

The results on the 50-dimensional functions are similar to those on the 30-dimensional functions. From Table 4, EO still ranks ninth and is only better than HFPSO. On the unimodal functions, MS-EO, Sa-DE and MSS-CS obtain one time ranking first, respectively. DS-PSO shows strong optimization ability on the multimodal and composition functions. However, MS-EO can obtain 6 times ranking first, and 13 times ranking the second. Moreover, the average ranking of MS-EO is 3.37 and it ranks first. It indicates that MS-EO also has the best optimization performance on the 50-dimensional function. Therefore, MS-EO shows stronger competitiveness on CEC2017 test set. In conclusion, the synthetization of many search strategies into EO is effective and MS-EO has stronger scalability and universality.

## DISCUSSION

### Statistical analysis

In order to verify MS-EO, this subsection performs Wilcoxon signed rank test with Bonferroni-Holm (*Zhang et al., 2020a*; *Zhang et al., 2020b*) correction based on the complex functions ($D = 30$ and $D = 50$) from CEC2017 test set (60 cases in sum). The purpose is to detect whether there is a significant difference between the two samples. Let H0 be the null hypothesis, that is, there is no significant difference between the two samples. The significance level ($b$) is set to 0.05 based on the best recommendation (*Derrac et al., 2011*). Table 5 provides Wilcoxon signed rank test results based on Table 3 and 4. $R^+$ represents the positive rank, which is the total rank of the problems in which MS-EO is better than the comparison algorithm. $R^-$ is the negative rank, which is the total rank of the problems in which MS-EO is inferior to the comparison algorithm. The corresponding rank is equally divided into $R^+$ and $R^-$ when the optimization performance of the two algorithms is equal. The $p$-value is calculated from $R^+$ and $R^-$. The Bonferroni-Holm method adjusts $b$ in descending order of $p$-value. $H_1$, $H_2$, …, $H_{k-1}$ are corresponding hypotheses. The hypothesis is rejected when $b/u < p$-value, otherwise it is accepted. $n/w/t/l$ respectively represent the number of optimization functions is $n$ and the number of functions that MS-EO is better than, equal to and inferior to the comparison algorithm.

From Table 5, the $p$-values of MS-EO and the comparison algorithms are all less than $b/u$, and the corresponding hypothesis is rejected. It shows that there are significant differences between MS-EO and the comparison algorithms. In particular, the function number that MS-EO is better than EO is 54. Moreover, the function number that MS-EO outperforms A-EO is 52. And MS-EO is significantly superior to any of the comparison algorithms on CEC2017 test set.

### Performance index analysis

In order to further evaluate MS-EO, this subsection uses the performance index ($P_i$) method (*Bharati, 1994*) to analyze and verify the optimization performance of MS-EO on CEC2017 test set. $P_i$ is a process of weighing the average error and running time. *Bharati (1994)* proposed $P_i$ to compare the relative performance of computational algorithms developed by her. This index gives a weighted importance to the success rate, the computational time as well as the number of function evaluations. Hence, it obtains a more accurate solution in

Sun et al. (2024), *PeerJ Comput. Sci.*, DOI 10.7717/peerj-cs.1760

**Table 4  Comparison results of MS-EO and 9 comparison algorithms on CEC2017 test set ($D = 50$).**

| Fun. | Metric | EO | DQL-SFLA | Sa-DE | ME-GWO | A-EO | HFPSO | HBBOG | DS-PSO | MSS-CS | MS-EO |
|------|--------|-----|----------|-------|--------|------|-------|-------|--------|--------|-------|
| $f_1$ | Mean | 3.9497E−03 | 2.3645E−02 | 2.5647E−03 | 2.1679E−04 | 3.2345E−03 | 8.3789E−03 | 2.3242E−03 | 2.9444E−03 | **4.3894E−01** | 1.6071E−01 |
| | Std | 3.5012E−03 | 4.0670E−02 | 3.1799E−03 | 6.0804E−03 | 2.9248E−03 | 9.2826E−03 | 1.9622E−03 | 3.2922E−03 | **1.1704E−00** | 2.6214E−01 |
| | Rank | 8 | 3 | 5 | 10 | 7 | 9 | 4 | 6 | **1** | 2 |
| $f_2$ | Mean | 1.6865E−18 | 4.5120E−40 | **1.8669E−08** | 5.6329E−18 | 2.8428E−37 | 1.5411E−13 | 6.1883E−13 | 1.2602E−16 | 3.9178E−08 | 2.7824E−11 |
| | Std | 8.0116E−18 | 2.2083E−41 | **7.9763E−08** | 2.5500E−19 | 1.7081E−38 | 9.7997E−13 | 3.0523E−14 | 7.9718E−16 | 2.1373E−09 | 9.8063E−11 |
| | Rank | 7 | 10 | **1** | 8 | 9 | 4 | 5 | 6 | 2 | 3 |
| $f_3$ | Mean | 3.0331E−03 | 5.3157E−04 | 1.8496E−03 | 8.7143E−03 | 1.1045E−04 | 1.8319E−01 | 6.5936E−02 | 1.5160E−01 | 1.1804E−04 | **7.3399E−07** |
| | Std | 2.2006E−03 | 9.6301E−04 | 1.5425E−03 | 3.0099E−03 | 4.0641E−03 | 5.9204E−01 | 3.9612E−02 | 2.2267E−01 | 3.9556E−03 | **2.6121E−06** |
| | Rank | 7 | 2 | 6 | 8 | 9 | 4 | 5 | 3 | 10 | **1** |
| $f_4$ | Mean | 8.4094E−01 | **1.6440E−00** | 1.0533E−02 | 4.0082E−01 | 9.1893E−01 | 8.5343E−01 | 3.5500E−01 | 2.3667E−02 | 4.1554E−01 | 5.8602E−01 |
| | Std | 4.5502E−01 | **3.2924E−00** | 4.3336E−01 | 1.8954E−01 | 5.5709E−01 | 5.1342E−01 | 1.8197E−01 | 3.0583E−01 | 3.7685E−01 | 4.5951E−01 |
| | Rank | 6 | **1** | 9 | 3 | 8 | 7 | 2 | 10 | 4 | 5 |
| $f_5$ | Mean | 1.4246E−02 | 1.7195E−02 | 1.2886E−02 | 1.4431E−02 | 1.4209E−02 | 1.7243E−02 | 1.2250E−02 | **3.4453E−01** | 1.2147E−02 | 6.4291E−01 |
| | Std | 2.6116E−01 | 3.8522E−01 | 2.3960E−01 | 2.1124E−01 | 3.3861E−01 | 4.1999E−01 | 1.4479E−01 | **7.2978E−00** | 1.4354E−01 | 1.3900E−01 |
| | Rank | 7 | 9 | 5 | 8 | 6 | 10 | 4 | **1** | 3 | 2 |
| $f_6$ | Mean | 6.2979E−01 | 2.5574E−01 | 5.3123E−01 | 2.8298E−01 | 4.0061E−02 | 4.4456E−00 | **1.6757E−05** | 1.2990E−03 | 3.6293E−02 | 8.7264E−04 |
| | Std | 6.1595E−01 | 3.4969E−00 | 5.8275E−01 | 6.1436E−02 | 5.0597E−02 | 5.3650E−00 | **7.7558E−06** | 4.3919E−03 | 2.6546E−02 | 2.5677E−03 |
| | Rank | 8 | 10 | 7 | 6 | 5 | 9 | **1** | 3 | 4 | 2 |
| $f_7$ | Mean | 1.9618E−02 | 2.8109E−02 | 2.4910E−02 | 1.8021E−02 | 1.9761E−02 | 2.0517E−02 | 1.6149E−02 | **7.3078E−01** | 1.7873E−02 | 1.0941E−02 |
| | Std | 3.5825E−01 | 4.4069E−01 | 4.8736E−01 | 1.8965E−01 | 3.4547E−01 | 3.2001E−01 | 1.6175E−01 | **4.2607E−00** | 1.8428E−01 | 2.2175E−01 |
| | Rank | 6 | 10 | 9 | 5 | 7 | 8 | 3 | **1** | 4 | 2 |
| $f_8$ | Mean | 1.4804E−02 | 1.7152E−02 | 1.3161E−02 | 1.4315E−02 | 1.4544E−02 | 1.7001E−02 | 1.2351E−02 | **3.3985E−01** | 1.2782E−02 | 7.1051E−01 |
| | Std | 2.9406E−01 | 3.6768E−01 | 2.0238E−01 | 1.8978E−01 | 3.2694E−01 | 4.5209E−01 | 1.0765E−01 | **8.1789E−00** | 1.5275E−01 | 1.2857E−01 |
| | Rank | 8 | 10 | 5 | 6 | 7 | 9 | 3 | **1** | 4 | 2 |
| $f_9$ | Mean | 3.5029E−02 | 3.6847E−03 | 1.1715E−03 | 1.9019E−02 | 4.1471E−02 | 1.7848E−03 | 7.7708E−01 | 6.6506E−01 | 1.3688E−02 | **4.0703E−01** |
| | Std | 3.4402E−02 | 1.0627E−03 | 6.8168E−02 | 1.7075E−02 | 6.5384E−02 | 2.3011E−03 | 1.6190E−02 | 8.1456E−01 | 9.7692E−01 | **5.1336E−01** |
| | Rank | 6 | 10 | 8 | 5 | 7 | 9 | 3 | 2 | 4 | **1** |
| $f_{10}$ | Mean | 5.9006E−03 | 5.7025E−03 | 4.7308E−03 | 4.9658E−03 | 6.1744E−03 | 5.3414E−03 | 3.7699E−03 | **3.2923E−03** | 4.1579E−03 | 8.6161E−03 |
| | Std | 1.0750E−03 | 7.0335E−02 | 7.3879E−02 | 5.5011E−02 | 1.3183E−03 | 8.6781E−02 | 3.7801E−02 | **6.7006E−02** | 5.1910E−02 | 1.1814E−03 |
| | Rank | 8 | 7 | 4 | 5 | 9 | 6 | 2 | **1** | 3 | 10 |

Sun et al. (2024), *PeerJ Comput. Sci.*, DOI 10.7717/peerj-cs.1760

**Table 4** (*continued*)

| Fun. | Metric | EO | DQL-SFLA | Sa-DE | ME-GWO | A-EO | HFPSO | HBBOG | DS-PSO | MSS-CS | MS-EO |
|------|--------|-----|----------|-------|--------|------|-------|-------|--------|--------|-------|
| | Mean | 1.3316E−02 | 1.1819E−02 | 1.2689E−02 | 8.1916E−01 | 1.0644E−02 | 2.0174E−02 | 8.3313E−01 | 1.4696E−02 | **5.2985E−01** | 5.8186E−01 |
| $f_{11}$ | Std | 5.1688E−01 | 3.1338E−01 | 3.5636E−01 | 1.3432E−01 | 3.4394E−01 | 5.7840E−01 | 1.9916E−01 | 2.8818E−01 | **8.7630E−00** | 1.2567E−01 |
| | Rank | 8 | 6 | 7 | 3 | 5 | 10 | 4 | 9 | **1** | 2 |
| | Mean | 8.7900E−05 | **3.6575E−03** | 7.4458E−05 | 1.8657E−05 | 2.1880E−06 | 1.4485E−06 | 2.4647E−05 | 3.5957E−05 | 6.9433E−05 | 3.8587E−03 |
| $f_{12}$ | Std | 6.1984E−05 | **2.2008E−03** | 4.2569E−05 | 5.4859E−04 | 1.3155E−06 | 1.6009E−06 | 1.0160E−05 | 4.3116E−05 | 5.2763E−05 | 2.7793E−03 |
| | Rank | 8 | **1** | 7 | 3 | 10 | 9 | 4 | 5 | 6 | 2 |
| | Mean | 6.7074E−03 | 4.1229E−03 | 2.5361E−03 | 6.6881E−02 | 1.1402E−04 | 2.3795E−04 | 3.9277E−03 | 3.1885E−03 | 2.9060E−02 | **2.9034E−02** |
| $f_{13}$ | Std | 7.1775E−03 | 1.5509E−03 | 2.5305E−03 | 6.7230E−01 | 9.6853E−03 | 2.5205E−04 | 3.1606E−03 | 4.1387E−03 | 6.6432E−01 | **7.7761E−01** |
| | Rank | 8 | 7 | 4 | 3 | 9 | 10 | 6 | 5 | 2 | **1** |
| | Mean | 5.3067E−04 | 1.0180E−02 | 4.7140E−04 | 1.5699E−02 | 9.8291E−04 | 3.5550E−04 | 1.1018E−03 | 1.1224E−04 | 7.8180E−01 | **5.5929E−01** |
| $f_{14}$ | Std | 4.7326E−04 | 1.7854E−01 | 3.1170E−04 | 1.5088E−01 | 7.0428E−04 | 3.1885E−04 | 8.7558E−02 | 8.6696E−03 | 1.5280E−01 | **1.0455E−01** |
| | Rank | 9 | 3 | 8 | 4 | 10 | 7 | 5 | 6 | 2 | **1** |
| | Mean | 1.1676E−04 | 1.0612E−03 | 5.6216E−03 | 1.4626E−02 | 1.4888E−04 | 1.1670E−04 | 1.4184E−03 | 2.3347E−03 | **4.7762E−01** | 8.0362E−01 |
| $f_{15}$ | Std | 6.9998E−03 | 2.7815E−02 | 4.7469E−03 | 1.6989E−01 | 6.0251E−03 | 1.1079E−04 | 1.0856E−03 | 3.6060E−03 | **7.5151E−00** | 2.0987E−01 |
| | Rank | 9 | 4 | 7 | 3 | 10 | 8 | 5 | 6 | **1** | 2 |
| | Mean | 1.1338E−03 | 1.1095E−03 | 1.0043E−03 | 9.8713E−02 | 8.9729E−02 | 1.2701E−03 | 1.0229E−03 | **5.3882E−02** | 1.0910E−03 | 1.2350E−03 |
| $f_{16}$ | Std | 3.6325E−02 | 2.5247E−02 | 3.3703E−02 | 1.7615E−02 | 3.8012E−02 | 3.7207E−02 | 2.3097E−02 | **1.9238E−02** | 2.7541E−02 | 4.2656E−02 |
| | Rank | 8 | 7 | 4 | 3 | 2 | 10 | 5 | **1** | 6 | 9 |
| | Mean | 9.6745E−02 | 8.7269E−02 | 7.5575E−02 | 6.0302E−02 | 7.5640E−02 | 9.3913E−02 | 6.6867E−02 | **4.8637E−02** | 7.3705E−02 | 9.9433E−02 |
| $f_{17}$ | Std | 3.2913E−02 | 2.0248E−02 | 1.8544E−02 | 1.3535E−02 | 2.5595E−02 | 2.6352E−02 | 1.9071E−02 | **1.4930E−02** | 1.3701E−02 | 2.3241E−02 |
| | Rank | 9 | 7 | 5 | 2 | 6 | 8 | 3 | **1** | 4 | 10 |
| | Mean | 3.2525E−05 | 7.5856E−02 | 5.5399E−05 | 1.1763E−03 | 1.2651E−06 | 1.5723E−05 | 3.5663E−04 | 6.8204E−04 | 4.8069E−04 | **4.7008E−01** |
| $f_{18}$ | Std | 2.3141E−05 | 3.3250E−02 | 4.5699E−05 | 3.4759E−02 | 6.5857E−05 | 1.4307E−05 | 1.4834E−04 | 3.3807E−04 | 2.0031E−04 | **8.9333E−00** |
| | Rank | 8 | 2 | 9 | 3 | 10 | 7 | 4 | 6 | 5 | **1** |
| | Mean | 1.4953E−04 | 3.0967E−02 | 1.3390E−04 | 6.6052E−01 | 1.8467E−04 | 2.0446E−04 | 2.7309E−02 | 1.3335E−04 | **3.0133E−01** | 3.2375E−01 |
| $f_{19}$ | Std | 1.0037E−04 | 1.5039E−02 | 8.3663E−03 | 7.4493E−00 | 1.2442E−04 | 1.9192E−04 | 9.4521E−01 | 6.3554E−03 | **5.7899E−00** | 4.3768E−00 |
| | Rank | 8 | 5 | 7 | 3 | 9 | 10 | 4 | 6 | **1** | 2 |
| | Mean | 6.8033E−02 | 6.2983E−02 | 5.6922E−02 | 4.6622E−02 | 4.8302E−02 | 6.9561E−02 | 5.4264E−02 | **2.3626E−02** | 5.3032E−02 | 8.0299E−02 |
| $f_{20}$ | Std | 2.6705E−02 | 1.5421E−02 | 1.4463E−02 | 1.1561E−02 | 2.4315E−02 | 2.9247E−02 | 1.6817E−02 | **1.1687E−02** | 1.8170E−02 | 2.4906E−02 |
| | Rank | 8 | 7 | 6 | 2 | 3 | 9 | 5 | **1** | 4 | 10 |
| | Mean | 3.1090E−02 | 3.3003E−02 | 3.1678E−02 | 3.3814E−02 | 3.1585E−02 | 3.7161E−02 | 3.3013E−02 | **2.3592E−02** | 3.2608E−02 | 2.6429E−02 |
| $f_{21}$ | Std | 2.8117E−01 | 2.3167E−01 | 2.3411E−01 | 2.1655E−01 | 2.3927E−01 | 3.2744E−01 | 1.3200E−01 | **6.5630E−00** | 1.5987E−01 | 1.6140E−01 |
| | Rank | 3 | 7 | 5 | 9 | 4 | 10 | 8 | **1** | 6 | 2 |

**Table 4** (*continued*)

| Fun. | Metric | EO | DQL-SFLA | Sa-DE | ME-GWO | A-EO | HFPSO | HBBOG | DS-PSO | MSS-CS | MS-EO |
|---|---|---|---|---|---|---|---|---|---|---|---|
| | Mean | 5.9913E−03 | 2.3991E−03 | 4.7690E−03 | 3.8077E−03 | 6.4205E−03 | 6.2153E−03 | 4.0848E−03 | **6.5559E−02** | 4.6340E−03 | 1.4748E−03 |
| $f_{22}$ | Std | 1.5479E−03 | 3.0495E−03 | 1.8229E−03 | 2.5873E−03 | 1.6431E−03 | 1.5579E−03 | 1.2150E−03 | **1.4155E−03** | 1.0493E−03 | 3.3287E−03 |
| | Rank | 8 | 3 | 7 | 4 | 10 | 9 | 5 | **1** | 6 | 2 |
| | Mean | 5.4238E−02 | 6.5156E−02 | 5.8431E−02 | 5.8174E−02 | 5.1925E−02 | 7.3856E−02 | 5.5405E−02 | **4.6855E−02** | 5.6194E−02 | 4.8880E−02 |
| $f_{23}$ | Std | 3.2819E−01 | 4.6161E−01 | 3.8846E−01 | 2.2401E−01 | 2.6376E−01 | 8.0298E−01 | 5.2356E−01 | **1.1961E−01** | 1.8753E−01 | 1.7623E−01 |
| | Rank | 4 | 9 | 8 | 7 | 3 | 10 | 5 | **1** | 6 | 2 |
| | Mean | 6.0057E−02 | 7.4405E−02 | 6.7632E−02 | 7.1514E−02 | 5.9131E−02 | 8.5766E−02 | 7.6565E−02 | **5.2496E−02** | 6.7982E−02 | 5.5065E−02 |
| $f_{24}$ | Std | 2.5696E−01 | 5.8372E−01 | 4.4248E−01 | 3.8843E−01 | 2.5283E−01 | 1.0176E−02 | 1.4692E−02 | **8.6792E−00** | 3.5212E−01 | 1.5233E−01 |
| | Rank | 4 | 8 | 5 | 7 | 3 | 10 | 9 | **1** | 6 | 2 |
| | Mean | 5.6300E−02 | **4.5604E−02** | 5.5431E−02 | 4.7884E−02 | 5.5861E−02 | 5.3195E−02 | 5.0183E−02 | 5.7977E−02 | 5.2177E−02 | 5.0583E−02 |
| $f_{25}$ | Std | 3.6744E−01 | **2.8919E−01** | 4.3335E−01 | 2.3321E−01 | 3.0101E−01 | 3.6107E−01 | 2.9610E−01 | 1.4104E−01 | 3.5843E−01 | 3.5559E−01 |
| | Rank | 9 | **1** | 7 | 2 | 8 | 6 | 3 | 10 | 5 | 4 |
| | Mean | 2.7101E−03 | 2.3459E−03 | 3.9842E−03 | 1.3313E−03 | 2.1844E−03 | 2.4549E−03 | 1.7512E−03 | **4.9789E−02** | 2.3641E−03 | 1.5420E−03 |
| $f_{26}$ | Std | 5.7969E−02 | 1.1079E−03 | 9.2623E−02 | 1.1968E−03 | 4.0895E−02 | 1.2922E−03 | 1.0885E−03 | **4.0639E−02** | 7.0239E−02 | 5.4648E−02 |
| | Rank | 9 | 6 | 10 | 2 | 5 | 8 | 4 | **1** | 7 | 3 |
| | Mean | 6.5759E−02 | 5.6750E−02 | 8.8692E−02 | 6.2301E−02 | 5.8586E−02 | 6.8377E−02 | 5.9493E−02 | 7.9031E−02 | 5.9110E−02 | **5.3873E−02** |
| $f_{27}$ | Std | 6.4481E−01 | 1.4337E−02 | 9.7798E−01 | 3.6098E−01 | 4.2581E−01 | 1.1189E−02 | 3.5636E−01 | 6.7411E−01 | 4.5603E−01 | **2.0866E−01** |
| | Rank | 7 | 2 | 10 | 6 | 3 | 8 | 5 | 9 | 4 | **1** |
| | Mean | 4.9836E−02 | **4.5767E−02** | 4.9924E−02 | 4.5958E−02 | 4.9536E−02 | 4.8912E−02 | 4.7089E−02 | 5.4023E−02 | 4.7078E−02 | 4.6787E−02 |
| $f_{28}$ | Std | 2.6521E−01 | **1.6478E−01** | 2.3252E−01 | 9.5307E−01 | 2.4678E−01 | 2.1819E−01 | 1.8699E−01 | 3.1154E−01 | 1.9905E−01 | 1.8554E−01 |
| | Rank | 8 | **1** | 9 | 2 | 7 | 6 | 5 | 10 | 4 | 3 |
| | Mean | 9.3754E−02 | 1.2909E−03 | 9.6535E−02 | 7.3576E−02 | 7.8757E−02 | 1.1964E−03 | 7.3439E−02 | **6.6524E−02** | 7.7557E−02 | 8.6297E−02 |
| $f_{29}$ | Std | 3.0896E−02 | 2.2162E−02 | 2.6606E−02 | 1.1266E−02 | 2.0488E−02 | 3.2623E−02 | 1.4813E−02 | **1.6706E−02** | 1.6510E−02 | 2.2128E−02 |
| | Rank | 7 | 10 | 8 | 3 | 5 | 9 | 2 | **1** | 4 | 6 |
| | Mean | 1.0765E−06 | **1.7409E−04** | 9.2269E−05 | 7.2462E−05 | 1.0418E−06 | 1.4015E−06 | 6.5634E−05 | 4.0551E−06 | 5.8299E−05 | 6.4818E−05 |
| $f_{30}$ | Std | 4.0254E−05 | **1.2783E−04** | 1.7675E−05 | 4.0653E−04 | 2.6907E−05 | 6.0428E−05 | 3.7787E−04 | 1.1743E−06 | 3.7944E−03 | 3.2330E−04 |
| | Rank | 8 | **1** | 6 | 5 | 7 | 9 | 4 | 10 | 2 | 3 |
| | Best/2nd best | 0/0 | 5/3 | 1/0 | 0/5 | 0/1 | 0/0 | 1/3 | 13/1 | 4/4 | 6/13 |
| | Ave. rank | 7.37 | 5.63 | 6.60 | 4.67 | 6.77 | 8.27 | 4.23 | 4.17 | 4.03 | 3.27 |
| | Total. rank | 9 | 6 | 7 | 5 | 8 | 10 | 4 | 3 | 2 | 1 |

**Table 5  Wilcoxon signed rank test with Bonferroni-Holm correction on the CEC2017 test set.**

|  | u | p-value | $R^+$ | $R^-$ | b/u | Hypothesis | n/w/t/l |
|---|---|---|---|---|---|---|---|
| MS-EO vs HFPSO | 9 | 5.7791E−09 | 1706 | 124 | 5.5556E−03 | Reject | 60/56/0/4 |
| MS-EO vs EO | 8 | 2.9685E−08 | 1668 | 162 | 6.2500E−03 | Reject | 60/54/0/6 |
| MS-EO vs A-EO | 7 | 9.8073E−07 | 1580 | 250 | 7.1429E−03 | Reject | 60/52/0/8 |
| MS-EO vs Sa-DE | 6 | 1.8000E−05 | 1498 | 332 | 8.3333E−03 | Reject | 60/50/0/10 |
| MS-EO vs DQL-SFLA | 5 | 3.2800E−04 | 1403 | 427 | 1.0000E−02 | Reject | 60/47/0/13 |
| MS-EO vs HBBOG | 4 | 1.2950E−03 | 1352 | 478 | 1.2500E−02 | Reject | 60/41/0/19 |
| MS-EO vs ME-GWO | 3 | 3.4720E−03 | 1312 | 518 | 1.6667E−02 | Reject | 60/43/0/17 |
| MS-EO vs DS-PSO | 2 | 1.4463E−02 | 1239 | 591 | 2.5000E−02 | Reject | 60/37/1/22 |
| MS-EO vs MSS-CS | 1 | 4.2174E−02 | 1191 | 639 | 5.0000E−02 | Reject | 60/39/0/21 |

**Notes.**
The bold values are set uniformly in order to correspond to the characters in Eqs. (1) and (12).

a fixed time if $P_i$ of the algorithm is higher, and its performance is better. The mathematical model of $P_i$ can be formulated by Eq. (19):

$$P_i = \frac{1}{m} \sum_{n=1}^{m} (c_1 \sigma_1^n + c_2 \sigma_2^n) \tag{19}$$

$$\sigma_1^n = M_E / E^m \tag{20}$$

$$\sigma_2^n = M_S / S^m \tag{21}$$

where $n \in [1, m]$, and $m$ is the sum of the number of problems. On the $n$-th problem, $M_E$ and $M_S$ are the minimum values of the average error value and average time of all algorithms, respectively. $E^m$ and $S^m$ the average error value and average time obtained by an algorithm on the $n$-th problem, respectively. $c_1$ is the average error weight and $c_2$ is the average time weight ($c_1 + c_2 = 1$ and $c_1, c_2 \in [0, 1]$).

Figure 9 shows the $P_i$ values of MS-EO and the contrast algorithms under the conditions of 30-dimensional and 50-dimensional functions in the CEC2017 test set. In addition, $m = 60$, the weight ($w$) is set to the numbers between 0 and 1 in steps of 0.2 according to the best recommendation from *Gupta et al. (2020)*, $c_1 = w$ and $c_2 = 1$-$w$. From Fig. 9, the value of $P_i$ of MS-EO is greater than the comparison algorithms at any time. Moreover, the value of $P_i$ of MS-EO is greater than that of EO and A-EO regardless of on CEC2017 test set. It shows that MS-EO can get more accurate solutions in a fixed time on CEC2017 test set. This further indicates that the performance of MS-EO is more stable.

## Convergence analysis

To validate the convergence quality of MS-EO, this subsection performs some convergence tests on the 30-dimensional functions from the CEC2017 test set. Four representative functions are selected from the CEC2017 test set to illustrate the problem briefly, namely, unimodal function $f_3$, multimodal function $f_9$, hybrid function $f_{12}$ and composition function $f_{30}$. Figure 10 illustrates the convergence curves of MS-EO and the comparison

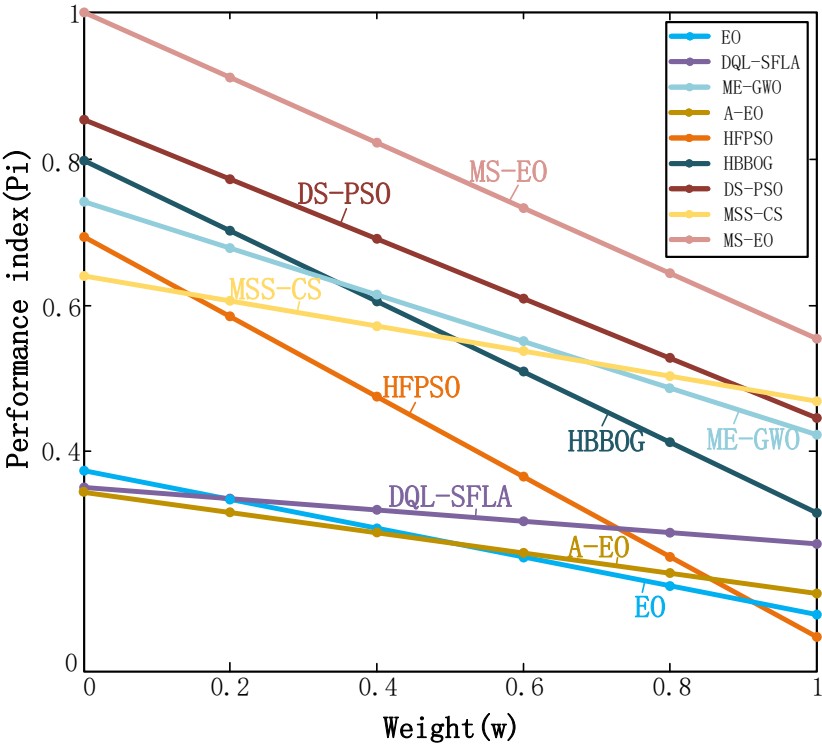

**Figure 9** Performance index curves on the CEC2017 test set.

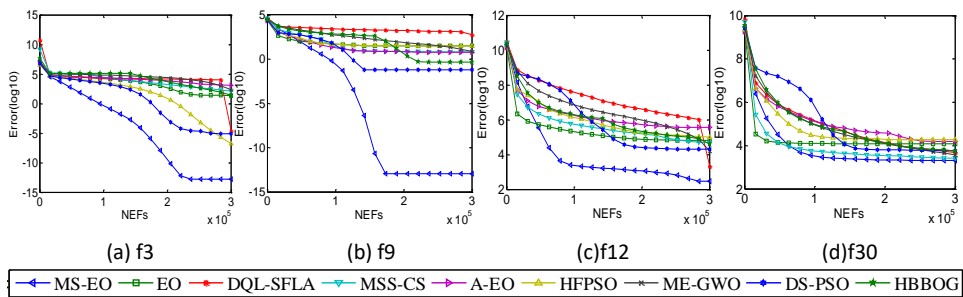

**Figure 10** Convergence curves on the 30-dimensional functions from the CEC2017 test set.

algorithms. From Fig. 10, the convergence speed of MS-EO is much faster than that of the comparison algorithms on $f_3$, $f_9$ and $f_{12}$.

On $f_{30}$, the convergence of MS-EO is not as obvious as on the first three functions. However, MS-EO converges faster than any of the comparison algorithms as the iteration number increases, and can find a more accurate solution. On $f_3$ and $f_{12}$, the convergence curve of DQL-SFLA drops suddenly and converges to a smaller solution in the late iteration stage. It is due to the fact that DQL-SFLA incorporates the quasi-Newton local search in the late stage, so that it has a faster convergence speed. In addition, the convergence speed

of MS-EO is dramatically faster than that of EO and A-EO on all four functions. MS-EO has better convergence performance compared with the comparison algorithms on the 30-dimensional functions from the CEC2017 test set.

## Application to feature selection

In order to verify the ability of MS-EO to solve practical problems, this section discusses MS-EO's application to feature selection. Feature selection is to find the optimal feature subset by eliminating irrelevant or redundant features from the original feature set (*Hu et al., 2021*). It can be defined as a discrete OP because the solution of feature selection belongs to binary. Suppose a data set has $N$ samples and $H$ features. $B$ is the original feature set. Then feature selection can be regarded as finding $h$ ($h < H$) features from $B$ while minimizing its fitness function to maximize the classification accuracy of a given classifier. Its search individual $X$ can be expressed as:

$$X_i^j = \begin{cases} 1 & C_i^j > 0.5 \\ 0 & \text{otherwise} \end{cases} \tag{22}$$

where 1 represents the corresponding feature is selected. Otherwise, it is not selected.

The objective function can be defined as:

$$\text{fitness} = \mu_1 \times e_r + \mu_2 \times (|h|/|H|) \tag{23}$$

where $\mu_1$ and $\mu_2$ are two weighting coefficients. $\mu_1 \in [0,1]$ and $\mu_2 = 1 - \mu_1$. According to the best recommendation (*Hu et al., 2021*), $\mu_1$ is set to 0.99, and the corresponding $\mu_2$ is 0.01. $e_r$ represents the error rate and it is calculated by $K$-Nearest Neighbor (KNN) classifier. KNN divides the dataset into training, validation, and testing of equal size to cross-verify each dataset (*Tu, Chen & Liu, 2019*). $K = 5$ in the experiment.

Twelve datasets from UCI machine learning repository (http://archive.ics.uci.edu/ml/index.php) we selected to test the effectiveness of MS-EO on feature selection. Table 6 provides a brief description of these datasets. $k$, $v$ and $z$ represent the number of samples, features and classifications of each dataset, respectively. Among them, the distribution of $D_7$ (Segmentation dataset) under different features is shown in Fig. 11. From Fig. 11, four plane graphs are made by selecting two of the 19 features of $D_7$. On Features 1 and 2, these samples with seven types fill the entire graph. In the remaining three feature plane graphs, the sample distribution is relatively concentrated because some sample points overlap.

Table 7 shows the comparison results of MS-EO and the comparison algorithms on the average fitness value of objective function (mean), classification accuracy (accuracy) and average number of selected features (AND). The common parameters are set as follows: For fair comparison, $M_{nfe} = 1000$ for all the comparison algorithms. For MS-EO, $N = 20$, $T = 50$; For ME-GWO, $N = 5$, $T = 100$; For SR-PSO, Sin-DE and WOASA, $N = 10$, $T = 100$. The experimental results of SR-PSO, Sin-DE, WOASA and ME-GWO are all taken from *Tu, Chen & Liu (2019)*.

From Table 7, MS-EO outperforms SR-PSO, Sin-DE and WOASA on all the datasets in terms of mean fitness. For ME-GWO, MS-EO outperforms it on all the remaining datasets except $D_4$. The classification accuracy is similar to mean. On accuracy, MS-EO

**Table 6  Brief description of the datasets.**

| No. | Name | k | v | z |
|-----|------|-----|-----|-----|
| $D_1$ | Australian | 690 | 14 | 2 |
| $D_2$ | Breast | 277 | 9 | 2 |
| $D_3$ | Hearts | 270 | 12 | 2 |
| $D_4$ | Ionosphere | 351 | 34 | 2 |
| $D_5$ | Kr_vs_kp | 3196 | 36 | 2 |
| $D_6$ | Sonar | 208 | 60 | 2 |
| $D_7$ | Segmentation | 210 | 19 | 7 |
| $D_8$ | Vowel | 528 | 10 | 2 |
| $D_9$ | Wine | 178 | 13 | 3 |
| $D_{10}$ | Waveform | 5000 | 21 | 3 |
| $D_{11}$ | Wdbc | 569 | 30 | 2 |
| $D_{12}$ | Zoo | 101 | 16 | 7 |

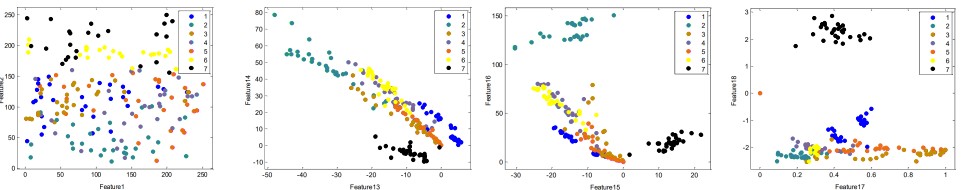

**Figure 11  Distribution graph of the Segmentation dataset ($D_7$) under different features.**

gets 11 times ranking the first, even reaching 100% on $D_{12}$. MS-EO gets the least number of features on all 10 data sets. On $D_4$, ME-GWO provides 95.91% classification accuracy by using 10.6 average features while MS-EO renders 95.68% accuracy with 9.6 average features. It suggests that MS-EO can effectively reduce the number of features. In order to highlight the comprehensive performance of MS-EO in solving feature selection problems, according to Table 7, the comprehensive ranking results of MS-EO and the comparison algorithms on feature selection are shown in Table 8. From Table 8, the ranking order of the five algorithms is MS-EO, ME-GWO, WOASA, Sin-DE and SR-PSO according to the comprehensive ranking results in Table 8. It further proves that MS-EO can solve the problem of feature selection more effectively.

## CONCLUSIONS

In order to cope with some drawbacks of EO in solving complex OPs, this article proposes a multi-strategy synthetized EO (MS-EO). Firstly, a simplified updating strategy is adopted by simplifying the original updating equation of EO to reduce the computational load and enhance the operability of EO. Secondly, an information sharing strategy based on iteration is added into the simplified EO. Most particles are updated by the strategy in the early search stage to enhance the global search ability while updated by the simplified EO in the late search stage ensuring the exploitation ability of EO. Then, a migration strategy is used

**Table 7  Comparison results based on mean, accuracy and AND.**

| | | Mean | | | | | Accuracy | | | | | AND | | | | |
|---|---|---|---|---|---|---|---|---|---|---|---|---|---|---|---|---|
| | | SR-PSO | Sin-DE | WOASA | ME-GWO | MS-EO | SR-PSO | Sin-DE | WOASA | ME-GWO | MS-EO | SR-PSO | Sin-DE | WOASA | ME-GWO | MS-EO |
| $D_1$ | | 1.9200E−01 | 1.4800E−01 | 1.3500E−01 | 1.2600E−01 | 1.2274E−01 | 80.93% | 85.39% | 86.67% | 87.54% | 87.83% | 4.6 | 4.8 | 4.2 | 3.2 | 3.1 |
| | Rank | 5 | 4 | 3 | 2 | 1 | 5 | 4 | 3 | 2 | 1 | 4 | 5 | 3 | 2 | 1 |
| $D_2$ | | 2.6400E−01 | 2.1100E−01 | 2.0600E−01 | 2.0400E−01 | 1.6826E−01 | 73.81% | 78.99% | 79.71% | 79.86% | 83.45% | 4.4 | 4.2 | 4.4 | 4.0 | 4.0 |
| | Rank | 5 | 4 | 3 | 2 | 1 | 5 | 4 | 3 | 2 | 1 | 4 | 3 | 4 | 1 | 1 |
| $D_3$ | | 2.4200E−01 | 1.8400E−01 | 1.5700E−01 | 1.4900E−01 | 1.4208E−01 | 76.00% | 81.78% | 85.19% | 85.33% | 85.93% | 4.8 | 4.6 | 4.2 | 4.0 | 3.3 |
| | Rank | 5 | 4 | 3 | 2 | 1 | 5 | 4 | 3 | 2 | 1 | 5 | 4 | 3 | 2 | 1 |
| $D_4$ | | 9.9000E−02 | 5.7000E−02 | 4.8000E−02 | 4.4000E−02 | 4.5574E−02 | 90.45% | 94.55% | 95.45% | 95.91% | 95.68% | 16.4 | 10.8 | 10.6 | 10.6 | 9.60 |
| | Rank | 5 | 4 | 3 | 1 | 2 | 5 | 4 | 3 | 1 | 2 | 5 | 4 | 2 | 2 | 1 |
| $D_5$ | | 2.2100E−01 | 1.9900E−01 | 1.8100E−01 | 1.7700E−01 | 1.7121E−01 | 78.20% | 80.23% | 82.17% | 82.53% | 83.09% | 18.6 | 16.4 | 15.2 | 15.2 | 13.5 |
| | Rank | 5 | 4 | 3 | 2 | 1 | 5 | 4 | 3 | 2 | 1 | 5 | 4 | 2 | 2 | 1 |
| $D_6$ | | 1.4200E−01 | 7.2000E−02 | 4.7000E−02 | 5.6000E−02 | 4.4247E−02 | 86.15% | 93.08% | 95.67% | 94.81% | 95.96% | 28.2 | 26.4 | 23.4 | 25.6 | 25.6 |
| | Rank | 5 | 4 | 2 | 3 | 1 | 5 | 4 | 2 | 3 | 1 | 5 | 4 | 1 | 2 | 2 |
| $D_7$ | | 1.1700E−01 | 8.4900E−02 | 8.9000E−02 | 5.5700E−02 | 5.0808E−02 | 88.76% | 91.81% | 91.43% | 94.67% | 95.14% | 9.8 | 7.0 | 8.0 | 5.2 | 4.9 |
| | Rank | 5 | 3 | 4 | 2 | 1 | 5 | 3 | 4 | 2 | 1 | 5 | 3 | 4 | 2 | 1 |
| $D_8$ | | 8.3000E−02 | 4.1000E−02 | 3.2000E−02 | 2.2000E−02 | 1.8250E−02 | 92.27% | 96.52% | 97.73% | 98.48% | 98.86% | 7.8 | 7.6 | 7.4 | 7.0 | 7.0 |
| | Rank | 5 | 4 | 3 | 2 | 1 | 5 | 4 | 3 | 2 | 1 | 5 | 4 | 3 | 1 | 1 |
| $D_9$ | | 8.1000E−02 | 4.8000E−02 | 2.8000E−02 | 1.9000E−02 | 1.5290E−02 | 92.36% | 95.53% | 97.51% | 98.43% | 98.88% | 6.6 | 4.6 | 4.2 | 4.0 | 5.0 |
| | Rank | 5 | 4 | 3 | 2 | 1 | 5 | 4 | 3 | 2 | 1 | 5 | 3 | 2 | 1 | 4 |
| $D_{10}$ | | 2.4800E−01 | 2.1500E−01 | 2.1200E−01 | 2.1700E−01 | 2.0721E−01 | 75.54% | 78.98% | 79.52% | 78.82% | 79.75% | 18.2 | 15.0 | 16.0 | 16.0 | 14.1 |
| | Rank | 5 | 3 | 2 | 4 | 1 | 5 | 3 | 2 | 4 | 1 | 5 | 2 | 3 | 3 | 1 |
| $D_{11}$ | | 8.2000E−02 | 7.3000E−02 | 6.1000E−02 | 5.6000E−02 | 4.9551E−02 | 92.21% | 92.77% | 94.04% | 94.53% | 95.12% | 5.2 | 5.4 | 5.2 | 5.0 | 3.8 |
| | Rank | 5 | 4 | 3 | 2 | 1 | 5 | 4 | 3 | 2 | 1 | 3 | 4 | 3 | 2 | 1 |
| $D_{12}$ | | 3.7000E−02 | 2.3000E−02 | 2.3000E−02 | 1.1000E−02 | 2.8125E−03 | 96.86% | 98.04% | 98.04% | 99.22% | 100% | 9.4 | 6.5 | 5.6 | 5.4 | 4.5 |
| | Rank | 5 | 3 | 3 | 2 | 1 | 5 | 3 | 3 | 2 | 1 | 5 | 4 | 3 | 2 | 1 |
| Count | | 0 | 0 | 0 | 1 | 11 | 0 | 0 | 0 | 1 | 11 | 0 | 0 | 1 | 3 | 10 |
| Ave. rank | | 5.00 | 3.75 | 2.92 | 2.17 | 1.08 | 5.00 | 3.75 | 2.92 | 2.17 | 1.08 | 4.67 | 3.75 | 2.75 | 1.83 | 1.33 |
| Total. rank | | 5 | 4 | 3 | 2 | 1 | 5 | 4 | 3 | 2 | 1 | 5 | 4 | 3 | 2 | 1 |

**Table 8  Comprehensive ranking results on feature selection.**

|  | SR-PSO | Sin-DE | WOASA | ME-GWO | MS-EO |
|---|---|---|---|---|---|
| Ave. Rank | 4.89 | 3.71 | 2.86 | 2.04 | 1.15 |
| Total. Rank | 5 | 4 | 3 | 2 | 1 |

for a golden particle to update and to enhance the search ability. Finally, an elite learning strategy is used for the worst particles in the late search stage to enhance the local search ability. Experimental results on CEC2013 and CEC2017 test sets demonstrate that MS-EO is better than EO and any of some state-of-the-art algorithms. MS-EO outperforms EO on 54 of 60 functions from CEC2017 test set ($D = 30$ and $D = 50$). The running time of MS-EO accounts for 36.22% and 45.45% of EO on the 30- and 50-dimensional functions, respectively. The value of $P_i$ of MS-EO is also the highest under any condition. On feature selection, MS-EO has also obtained more convincing results. These results on both complex functions and feature selection show many updating strategies synthesized into EO are effective and expect to be extended to improvement on other MAs. In the future works, we will propose new search strategies to further improve the multi-strategy theory for MS-EO and other MAs. The improved MAs will be dealt with more practical problems.

### Funding

This work was supported by the Henan Province Soft Science Research Plan Projects (No. 212400410109), the Henan Province Science Foundation for Youths (No. 222300420058), the National Natural Science Foundation of China under Grant (No. 62002103), the Science and Technology Research Project of Henan Provincial Science and Technology Department (No. 232102321064) and the 2021 Henan Province higher Education Teaching Reform research and practice key project (No. 2021SJGLX320). The funders had no role in study design, data collection and analysis, decision to publish, or preparation of the manuscript.

### Grant Disclosures

The following grant information was disclosed by the authors:
Henan Province Soft Science Research Plan Projects: 212400410109.
Henan Province Science Foundation for Youths: 222300420058.
National Natural Science Foundation of China: 62002103.
Science and Technology Research Project of Henan Provincial Science and Technology Department: 232102321064.
2021 Henan Province higher Education Teaching Reform research and practice key project: 2021SJGLX320.

### Competing Interests

The authors declare there are no competing interests.

## Author Contributions

- Quandang Sun conceived and designed the experiments, authored or reviewed drafts of the article, and approved the final draft.
- Xinyu Zhang conceived and designed the experiments, authored or reviewed drafts of the article, and approved the final draft.
- Ruixia Jin performed the experiments, authored or reviewed drafts of the article, and approved the final draft.
- Xinming Zhang analyzed the data, performed the computation work, prepared figures and/or tables, and approved the final draft.
- Yuanyuan Ma conceived and designed the experiments, prepared figures and/or tables, and approved the final draft.

## Data Availability

The data are available at GitHub:

- CEC2013: https://github.com/P-N-Suganthan/CEC2013
- CEC2017: https://github.com/P-N-Suganthan/CEC2017.

## Supplemental Information

Supplemental information for this article can be found online at http://dx.doi.org/10.7717/peerj-cs.1760#supplemental-information.

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
