# Peer review of "Multi-strategy synthetized equilibrium optimizer and application"

_PeerJ Computer Science, doi:10.7717/peerj-cs.1760_

## Round 0.1 · original submission · Minor Revisions

Dear authors,

Thank you for your submission. Your article has not been recommended for publication in its current form. However, we do encourage you to address the concerns and criticisms of the reviewers and resubmit your article once you have updated it accordingly. Reviewer 3 has requested that you cite specific references. You may add them if you believe they are especially relevant. However, I do not expect you to include these citations, and if you do not include them, this will not influence my decision.

Best wishes,

**Language Note:** PeerJ staff have identified that the English language needs to be improved. When you prepare your next revision, please either (i) have a colleague who is proficient in English and familiar with the subject matter review your manuscript, or (ii) contact a professional editing service to review your manuscript. PeerJ can provide language editing services - you can contact us at copyediting@peerj.com for pricing (be sure to provide your manuscript number and title). – PeerJ Staff

Reviewer 1 ·

Basic reporting

1- The related works section should provide more detailed information about the elite group.
2-More comprehensive explanations regarding the flow chart of MS-EO within the context of the proposed algorithm.

Experimental design

'no comment'

Validity of the findings

1-In the results section, it would be beneficial to discuss the calls made to the objective function and their impact on the algorithm's convergence.

Additional comments

The article presents an engaging topic and articulates a compelling problem statement.However, it would be advantageous to delve into the impact of noise on the algorithm and explore the constraints of the proposed method.

Cite this review as

·

Basic reporting

'no comment'

Experimental design

'no comment'

Validity of the findings

'no comment'

Additional comments

Dear Author,

Thank you for your submission. Below are some of my comments and feedback after reading the abstract and conclusion sections of your paper:

Clear and Concise Introduction: The abstract starts with a concise background, highlighting the problems associated with equilibrium optimizer (EO). This establishes a solid foundation for the reader to appreciate the improvements you introduce in MS-EO.

Methodological Advancements: I appreciate the step-by-step introduction of the different strategies you implemented in MS-EO. The progression from SS-EO to GS-EO and finally MS-EO showcases the iterative nature of your improvements.

Balancing Exploration and Exploitation: The manner in which you've embedded various strategies to strike a balance between exploration and exploitation is commendable. However, it might be helpful if you could elaborate a bit more on the mechanisms that help in achieving this balance, possibly in the main content.

Results Section: It's commendable that your experimental results are comprehensive, covering comparisons with state-of-the-art algorithms and tests on multiple datasets. This strengthens the credibility of your proposed method.

Conclusions: Your conclusion summarizes the paper's content effectively. It reiterates the main strategies implemented and their respective advantages, aiding the reader in recapitulating the paper's significant points.

Terminology Consistency: Ensure consistency in naming conventions. For example, if you refer to "equilibrium optimizer" as "EO", it should be consistently referred to as such throughout the paper.

Future Works: It's promising that you're looking to expand on this work further. It would be interesting to see how new search strategies can elevate the capabilities of MS-EO and other metaheuristic algorithms (MAs).

Potential Improvement:

Consider using graphics or flowcharts in the main content to visually represent the transformation from EO to MS-EO. This can help readers, especially those less familiar with the domain, to grasp the concept more quickly.
Elaborate a bit more on the "value of Pi." Its significance and why it's worth mentioning as a metric of comparison.
Recommendation for Clarity: While the abstract is comprehensive, consider breaking it into smaller paragraphs for easier reading. This might make it more digestible, especially for readers who are skimming.

Final Thoughts: Overall, the paper seems promising in addressing the mentioned drawbacks of EO. If the main content is as detailed and thorough as the abstract and conclusion, I expect this to be a valuable contribution to the field.

I look forward to reviewing the complete paper and understanding the depth of your proposed strategies and methodologies.

Warm Regards,

Cite this review as

Reviewer 3 ·

Basic reporting

I think this part is well presented. Although there are so many new algorithms out there that the authors better have a brief overview of these new algorithms. As:
[A]"A new metaphor-less simple algorithm based on Rao algorithms: a Fully Informed Search Algorithm (FISA)." PeerJ Computer Science 9 (2023): e1431.

[B] "Geyser Inspired Algorithm: A New Geological-inspired Meta-heuristic for Real-parameter and Constrained Engineering Optimization." Journal of Bionic Engineering (2023): 1-35.

Experimental design

In my opinion, it is better for the authors to test the proposed method on various engineering problems from the real world.

Validity of the findings

In my opinion, the proposed method and the results obtained and reported are valuable and significant by researchers in this field.

Additional comments

In general, the proposed method is suitable and worthy of attention, although there is still a need for improvements in the presented article.

Cite this review as

---

## Round 0.2 · accepted · Accept

Dear authors,

Thank you for the revision and for clearly addressing all the reviewers' comments. The paper seems to be improved in the opinion of the reviewers. Your article seems to be acceptable for publication after the last revision.

Best wishes,

Reviewer 1 ·

Basic reporting

'no comment'

Experimental design

'no comment'

Validity of the findings

'no comment'

Additional comments

The authors’ response to the noise discussion is convincing, although it would have been better to use a filter to eliminate the noise effect.

Cite this review as

·

Basic reporting

done

Experimental design

done

Validity of the findings

done

Additional comments

no

Cite this review as